# Long term trends in diatom diversity and palaeoproductivity: a 16,000-year multidecadal study from Lake Baikal, southern Siberia

**Anson W. Mackay[1*], Vivian A. Felde[2], David W. Morley[1], Natalia Piotrowska[3], Patrick Rioual[4], Alistair W.R. Seddon[2], George E.A. Swann[5]**

[1]Anson W. Mackay*
Environmental Change Research Centre, Department of Geography, UCL, London UK, WC1E 6BT.
ans.mackay@ucl.ac.uk
*Corresponding Author

[2]Vivian Astrup Felde
Department of Biological Sciences, and Bjerknes Centre of Climate Research, University of Bergen, PO Box 7803, Bergen N-5020, Norway
Vivian.Felde@uib.no

[1]David W. Morley
Environmental Change Research Centre, Department of Geography, UCL, London UK, WC1E 6BT.
d.w.morley@gmail.com

[3]Natalia Piotrowska
Department of Radioisotopes, Institute of Physics - CSE, Silesian University of Technology, Konarskiego 22B, 44-100 Gliwice, Poland
Natalia.Piotrowska@polsl.pl

[4]Patrick Rioual
Key Laboratory of Cenozoic Geology and Environment, Institute of Geology & Geophysics, Chinese Academy of Sciences, P.O. box 9825, Beijing 100029, China
prioual@mail.igcas.ac.cn

[2]Alistair W. R. Seddon
Department of Biological Sciences, and Bjerknes Centre of Climate Research, University of Bergen, PO Box 7803, Bergen N-5020, Norway
alistair.seddon@uib.no

[5]George E. A. Swann
School of Geography, University of Nottingham, University Park, Nottingham, NG7 2RD, UK
George.Swann@nottingham.ac.uk

Correspondence to: Anson W. Mackay (ans.mackay@ucl.ac.uk)

**Abstract**

Biological diversity is inextricably linked to community stability and ecosystem functioning,
but our understanding of these relationships in freshwater ecosystems is largely based on
short-term observational, experimental, and modelling approaches. Using a multidecadal
diatom record for the past c. 16,000 years from Lake Baikal, we investigate how diversity
and palaeoproductivity have responded to climate change during periods of both rapid
climate fluctuation, and relative climate stability. Here we show dynamic changes in diatom
communities during the past 16,000 years, with decadal shifts in species dominance
punctuating millennial-scale seasonal trends. We describe for the first time in Lake Baikal a
gradual shift from spring to autumnal diatom communities that started during the Younger
Dryas and peaked during the Late Holocene, and likely represent orbitally-driven ecosystem
responses to long-term changes in seasonality. Using a multivariate classification tree, we
show that trends in planktonic and tychoplanktonic diatoms broadly reflect both long-term
climatic changes associated with the demise of Northern Hemisphere ice sheets, and abrupt
climatic changes associated with, for example, the Younger Dryas stadial. Indeed, diatom
communities are most different before and after the boundary between the Early and Middle
Holocene periods c. 8.2 cal kyr BP, associated with the presence and demise of northern
hemisphere ice sheets respectively. Diatom richness and diversity, estimated using Hill's
species numbers, are also shown to be very responsive to periods characterised by abrupt
climate change, and using knowledge of diatom autecologies in Lake Baikal, diversity trends
are interpreted in terms of resource availability. Using diatom biovolume accumulation rates
(BVAR; $\mu m^3$ $cm^{-2}$ $yr^{-1}$), we show that spring diatom crops dominate palaeoproductivity for
nearly all of our record, apart from a short period during the late Holocene, when autumnal
productivity dominated between 1.8–1.4 cal kyr BP. Palaeoproductivity was especially
unstable during the Younger Dryas, reaching peak rates of 18.3 x $10^3$ $\mu m^3$ $cm^{-2}$ $yr^{-1}$ at c. 12.3
cal kyr BP. Generalized additive models (GAM) to explore productivity–diversity
relationships (PDR) during pre-defined climate periods, reveal complex relationships.
Strongest statistical evidence for GAMs were found during the Younger Dryas, the Early
Holocene and the Late Holocene, i.e. periods of rapid climate change. We account for these
differences in terms of climate-mediated resource availability, and the ability of endemic
diatom species in Lake Baikal to adapt to extreme forms of living in this unique ecosystem.
Our analyses offer insight into how productivity–diversity relationships may develop in the
future under a warming climate.

**Key words:** palaeoproductivity, abrupt climate change, palaeolimnology, Holocene, Late
glacial, productivity–diversity relationship, generalized additive models, multivariate
classification trees




## 1. Introduction


Understanding the role that biological diversity plays in ecosystem stability and function is an
important challenge in ecological research (Tilman et al. 1997; McCann 2000; Loreau et al.
2001; Isbell et al. 2015; Hagen et al. 2021). An ecosystem with higher biodiversity is
assumed to be more stable, due to a number of factors, including the presence of species
which have considerable plasticity, such that they have wide responses to disturbances
(McCann 2000; Luethje and Snyder 2021), and the "insurance effect" (Yachi and Loreau
1999) where species redundancy plays an important stabilizing role. Biodiversity loss can
lead to reduced ecosystem functioning (Cardinale et al. 2012), which poses serious threats to
ecosystem health in general, and human well-being in particular (e.g. Chivian 2003). While
the mechanisms behind how diversity controls functioning are not completely understood,
dominant species with distinct functional roles are important (Winfree et al. 2015). For
example, primary production, a key ecosystem function which helps regulate the global
carbon cycle, is tied to species diversity (Tilman et al. 1997; Cardinale et al. 2009; Tilman et
al. 2012). However, our understanding of productivity–diversity relationships (PDR) is
largely based on short-term observational (Dodson et al. 2000; Ptacnik et al. 2008; Korhonen
et al. 2011), experimental (McGrady-Steed et al. 1997; Interlandi & Kilham 2001; Winfree et
al. 2015) and modelling approaches (Aoki 2003), with very few long-term studies undertaken
(Rusak et al. 2004). This constitutes an important gap in our knowledge because in terms of
climate change, productivity–diversity relationships and resource use efficiency (Gross and
Cardinale 2007; Ptacnik et al. 2008) will be fundamentally different over long (e.g. climate
and landscape evolution) and short (e.g. pulse disturbances such as climate disturbance
events (Kéfi et al. 2019)) timescales.

Palaeoecological approaches provide a potential solution to this challenge, since they can
reveal ecological dynamics in response to environmental change which unfold only over very
long timescales (National Research Council 2005; Seddon et al. 2014). For example, diatoms
are some of the most important primary producers in lake ecosystems, and their sub-
fossilised remains provide a record of community variations through time. In addition,
because the species composition, biovolume and concentration of diatom valves can be
directly measured on sediment sequences, there is a unique opportunity to investigate how
community dynamics, diversity and the stability of ecosystem functions such as productivity
change over long-timescales. In particular, because the last 16,000 years have been
punctuated by a series of large-scale and abrupt climatic shifts, such records open the door
for an investigation into the links between biodiversity and the stability of ecosystem
functioning associated with climate disturbance events.

Here we investigate the relationship between diatom diversity and ecosystem functioning in
Lake Baikal, an ancient lake with a long continuous record with evidence of only very recent
human perturbation (Izmest'eva et al. 2016; Roberts et al. 2018) restricted to its coastline
(Kravtsova et al. 2014; Timoshkin et al. 2016). We focus on aquatic productivity as a
measure of ecosystem function, because of the direct link between diatoms and primary
production in the modern lake (Kozhova and Izmest'eva 1998).

As yet we do not assume to know the precise nature of the potential productivity–diversity
relationships that may have occurred in Lake Baikal over such long timescales. We do know
however, that local diversity will be influenced by a range of chemical, biological and
physical factors such as nutrient availability, species interactions, and disturbance events such
as rapid climate change.  Our approach therefore is to explore productivity– diversity
relationships over specific timescales independently determined from palaeoclimate studies.
We might hypothesise for example, that productivity and diversity will co-vary linearly
during rapid warming because increased energy results in increased metabolic rates in the
system, but that this relationship might change during periods of relative climate stability. We
investigate the range of possible response functions based on those identified by Smith
(2007), whether they be negative or positive, whether they be humped or U-shaped, flat or
just random (Smith 2007).

One of the most relevant independent climate studies for this time period is by Tarasov et al.
(2009), who modelled pollen-inferred palaeotemperature for neighbouring Lake Kotokel
(Fig. 1) for the past 15,000 years. Their reconstructions indicate a period of rapid warming
(during the Bølling–Allerød interstadial), rapid cooling (during the Younger Dryas stadial)
and relative temperature stability of the Holocene in southern Siberia. However, given that
the Holocene has recently been formally subdivided into three ages/subepochs (Walker et al.
2018),  it is possible to compare productivity– diversity relationships during the Younger
Dyras and the Bølling–Allerød, with the Greenlandian/Early, Northgrippian/Middle, and
Meghalayan/Late ages/subepochs of the Holocene (henceforth referred to as Early, Middle,
and Late Holocene). Our approach is to coax history to conduct experiments (Deevey 1969)
in order to deepen our understanding of (very) long-term biodiversity–ecosystem functioning
(Loreau 2001) through the exploration of relationships between diatom diversity and
productivity, and to test hypotheses related to how rapid climate change may disrupt these
relationships on sub-orbital timescales. We do this by:
• reconstructing trends in diatom communities, diatom diversity and palaeoproductivity
in Lake Baikal at a multidecadal resolution for the past c. 16,000 years.
• hypothesizing that relationships between productivity–diversity will differ during
periods of rapid climate change and periods of relative climate stability.

**2. Methods**

Study site
Lake Baikal is situated in southern Siberia at the forest – steppe ecotone (Fig. 1), and is the
world's oldest, deepest, and most voluminous lake. The lake is a World Heritage Site, due to
its diverse flora and fauna; it contains over 2,500 species of which over 75% are thought to
be endemic (Galazii 1989). Its sediments have accumulated for over 25 million years, and
decades of research have exploited this sedimentary record to reconstruct environmental
change (Williams et al. 2001; Mackay 2007). Sedimentary cores were obtained in 2001 from
the research vessel *Vereschagin*, from the Vydrino Shoulder (51.585° N, 104.855° E; water
depth 675 m), an underwater high (between 500 – 800 m) off the south-eastern coast of Lake
Baikal (Fig. 1). The Vydrino Shoulder was selected because it is an area of stable, fine-
grained sedimentation relatively undisturbed by tectonic activity and reworking (Charlet et al.
2005). The cores obtained included a 1.73 m trigger core (CON01–605–3a), a 10.45 m piston
core (CON01–605–3), and a 2.50 m box core (CON01–605–5). These records were
supplemented with a short gravity core taken from an ice platform in 2001 (CON01–105–6)
from the same region.

Dating
Radiocarbon dates for the Vydrino sequence were obtained by accelerated mass spectrometry
(AMS) from pollen and spore concentrates (Piotrowska et al. 2004; Demske et al. 2005). The
age model is based upon twelve AMS [14]C pollen dates from the box core (CON01–605–5)
(Piotrowska et al. 2004) and an additional five AMS [14]C pollen dates from the piston core
(CON01–605–3) (Demske et al. 2005). Full details are given in Mackay et al. (2011).
Radiocarbon dates were calibrated using IntCal20 radiocarbon calibration curve (Reimer *et*
*al.*, 2020), and age-depth modelling was done using 'Bacon2.2' (Blaauw & Christen, 2011).
No reservoir effect was applied to the calibration because the organic material used for
dating, i.e. pollen and spore concentrates, is assumed to be uncontaminated by old carbon.

Diatom analysis
Diatoms amount to between 50-90% of the phytoplankton biomass during spring bloom
under ice and after ice break-up (Popovskaya et al. 2015; Panizzo et al. 2017). Spring bloom
contributes a significant proportion of overall annual primary productivity (Popovskaya
2000). With the onset of summer warming and surface water stratification, diatoms are
succeeded by non-siliceous autotrophic picoplankton and other green algae (Fietz et al. 2005;
Belykh et al. 2006). During autumn turnover, a smaller diatom bloom dominates primary
production. Nitrogen and phosphorus co-limit photic zone productivity in Lake Baikal (Satoh
et al. 2006; O'Donnell et al. 2017), with rates of deep-water nutrient supply increasing
markedly since the mid 19th century (Swann et al. 2020).

Diatoms are siliceous, so they generally preserve well in sedimentary environments. We
prepared diatom samples for microscopy from sediments sampled every 5mm from the
composite sequence derived from the  gravity / trigger / piston cores detailed above. Unlike
standard diatom preparation analyses (Battarbee et al. 2001), no chemical treatments were
needed, although we enabled diatom concentrations to be calculated through the addition of
divinylbenzene microspheres (Mackay et al. 1998). Diatom taxa were identified to species
level or lower using a range of Russian and other floras, detailed in Mackay et al. (1998).
With few exceptions, at least 300 valves from each sample were counted using oil immersion
phase-contrast light microscopy at x1000 magnification. Planktonic and tychoplanktonic
diatoms account for on average, c. 90 % of all diatoms counted, and here we detail diatom
compositional change for planktonic and tychoplanktonic species only, as these were used to
determine palaeoproductivity estimates below. The diatom taxonomical nomenclature was
updated according to the global online database AlgaeBase https://www.algaebase.org/.

Detrended correspondence analysis (DCA) and canonical correspondence analysis (CCA)
were used to investigate the total amount of turnover, variation, and compositional changes of
the diatom assemblages over time. The analyses were performed on Hellinger transformed
data to dampen the effects of highly abundant taxa (Legendre and Gallagher 2001).
Multivariate classification tree (MCT) analyses was used to explore how much of the
variation the different climatic periods identified above can be explained by the diatom
assemblages. MCT is considered a powerful method when there is a non-linear relationship
between response and explanatory variables, when there are missing values in the data, or
there are higher level interactions between explanatory variables (Borcard et al. 2018). The
minimum sized tree was chosen based on the 1se rule, which is the minimum sized tree when
the cross-validated relative error (CVRE) is the minimum CVRE value plus 1 standard error
(SE) of all the CVRE values (Borcard et al. 2018). The number of multiple cross-validations
was set to 100, and the number of $k$ groups was set to the number of rows in the dataset. The
counts were normalized prior to the analysis and since MCT is in Euclidean space it
transforms into the chord distances (sensu the original chord distances proposed by Orloci
1967) between samples. Ordination analyses were done using the R package vegan and
results plotted using ggvegan. The MCT was done using the R package mvpart.
Stratigraphical profiles were constructed using C2 Data Analysis Version 1.7.7 (Juggins

239    2014).


Palaeoproductivity
The palaeoecological significance of diatom concentrations can be of limited value unless
they can be converted into either diatom flux rates (e.g. Battarbee et al. 2001), or estimates of
biovolume (Hillebrand et al., 1999). Diatom flux rates rely on a robust chronology (as
determined in this study), while biovolume estimates require representative size
measurements for the dominant taxa. We estimate palaeoproductivity through community
biomass calculations derived from diatom biovolume accumulation rates (BVAR; $\mu m^3$ $cm^{-2}$
$yr^{-1}$) for all major planktonic and tychoplanktonic species (Interlandi and Kilham 2001).
Biovolumes ($\mu m^3$) were calculated using the median of measured linear dimensions of 25
valves, and geometric shape guidelines (Hillebrand et al. 1999). Species used for biovolume
calculations accounted for on average, over 80 % of total relative abundance, and therefore
we assume are a robust estimate of palaeoproductivity.

Diatom richness, diversity and evenness
Richness, diversity and evenness were estimated using Hill's species numbers $N0$, $N1$, and
$N2$ using diatom count data of the planktonic and tychoplanktonic flora only. We do not
include benthic taxa because they represent a different habitat, far from the core location.
Their persistent presence in the core (average 10 %) is caused by secondary transport
processes rather than species competing for the same resources as the planktonic diatoms
themselves.  Hill's species numbers give easily interpretable numbers by maintaining the
doubling effect, and provide information on three levels as to how rare and abundant taxa are
weighted in each sample (Hill, 1973; Jost, 2010; Gotelli and Ellison, 2013; Chao et al., 2014).
Species richness is estimated using Hill's $N0$, the expected number of taxa where rare and
abundant taxa have similar weights. Species diversity is estimated using Hill´s $N1$, which is
the expected number of equally common taxa, with less weight on  rare taxa, and $N2$, which
provides the expected number of equally abundant taxa but puts weight on the numerically
dominant taxa. The estimates are represented as the expected number of diatoms based on the
smallest sample size (n = 150) to avoid biases related to different sample sizes. To take
account of variable sediment accumulation rates (SAR), Hill's measures of richness and
diversity were further divided by accumulation rates over time, so these measures become
estimated Hill's numbers per $cm^2$ $yr^{-1}$. To detect variation in abundance changes over time
(evenness) we also include the ratio of $N2/N1$ (Birks et al. 2018), i.e. the proportion of very
abundant species to the number of common species identified. When the ratio is 1, it
indicates that all taxa were equally abundant. The ratio is calculated based on the Hill's
numbers after taking account of SAR.

Palaeoproductivity – Diversity relationships
Relationships between palaeoproductivity and $N2$ diversity were investigated during pre–
identified time–intervals (Tarasov et al. 2009; Walker et al. 2018) to test the hypothesis that
PDR will differ during periods of rapid climate change and periods of relative climate
stability. Defined climate periods are the pre– Bølling–Allerød (15.9–14.7 cal kyr BP), the
Bølling–Allerød (14.7–12.9 cal kyr BP), the Younger Dryas (12.9–11.7 cal kyr BP), and the
three recently designated sub-epochs of the Holocene (Early (11.7–8.2 cal kyr BP), Middle
(8.2–4.2 cal kyr BP) and Late (4.2–0 cal kyr BP). We used generalized additive models
(GAM) to explore PDR because we have no prior knowledge of the expected relationships,
and the likelihood that the relationships were non-linear or showing different complex
patterns within the different time periods. We used $N2$ diversity as the response variable and
diatom BVAR as the explanatory variable. $N2$ is a continuous variable so we used a gamma
distribution with a log link. Diatom BVAR as a variable was skewed so it was log
transformed prior to the analyses, and climatic period was included as a factor variable that
allowed for different smooths for each period. The smooths were fitted using thin-plate
regression splines, and the model was fitted using the restricted maximum likelihood
approach. Prior to analyses, extreme outliers ($BVAR_{log}$ $\mu m^3$ $cm^{-2}$ $yr^{-1}$) < 6 were removed to
reveal more clearly GAM relationships. This led to only 8 out of 521 samples being removed
in total: three, two and one samples removed from the Late, Middle and Early Holocene
subepochs respectively, and a further two from the pre–BA period. Models were then refitted.

The model is specified as follows:

$\log([N2_{ij}]) = a_1 + f_1(Productivity_i, Period_j) + \varepsilon_i \quad N2_i \sim Gamma$

where $i$ represents each sample, $j$ is each climate period and $a_1$ is the intercept term for the
periods. Since the data are time ordered, the data points may not be independent and we
added a correlation term to the model attempting to reduce effects of potential temporal
autocorrelation. However, adding different correlation structures did not improve the model
or affect the residual variation, and we therefore removed them to keep the simplest model.

**3. Results**

Chronology
The calibrated ages for our profile span the last 15.91 cal kyr BP (where kyr = thousands of
years, BP = before present, i.e. 1950 CE (common era)) (Figure 2). Sediment accumulation
rates ranged between 32–184 yr $cm^{-1}$ (mean 62 yr $cm^{-1}$).

Diatoms
The temporal resolution of species compositional change was c. 30 years for the past c.
16,000 years (Fig. 3). DCA axis 1 sample scores show a strong, largely unidirectional trend
since the start of the Holocene. Turnover in the planktonic/tychoplanktonic assemblages over
the whole time period is 2.71 standard deviation units, reflecting the observation that several
species present during the early part of our record are also present during the latter part of our
record. The total inertia (variation) measured by CCA is 1.82, and of this the constraining
time variable explains 10.6 % of the variation (eigenvalue = 0.193). The ratio of eigenvalues
of the constrained axis 1 over the unconstrained axis 2 is 1.29 indicating that the time
gradient is explaining an important component of the variation.

During the early stages of the pre– Bølling–Allerød period, $N0$ and $N2$ values were initially
relatively high (Fig. 4), before declining as the tychoplanktonic *Aulacoseira skvortzowii*
dominated the assemblage (Fig. 3). Dominance by *A. skvortzowii* persists into the Bølling,
but is gradually reduced by increasing abundances of the planktonic *Aulacoseira baicalensis*
during the later Allerød. DCA axis 1 ordination scores show very little change during the
pre– Bølling–Allerød and Bølling–Allerød interstadial, indicative of a rather stable
assemblage, with few equally abundant species reflected in low evenness scores ($N2/N1$).
MCT is not able to discriminate well between diatom assemblages of the pre-Bølling–Allerød
and Bølling–Allerød interstadial (Fig. 5).

The Younger Dryas stadial is characterised by major changes in the diatom assemblage –
abundances of *Aulacoseira* (especially *A. skvortzowii*) decline, concomitant with a rapid
succession of *Ulnaria acus* and *Crateriportula inconspicua*, that bloom in spring and autumn
respectively (Fig. 3). These taxa give way to a short-lived peak in *A. baicalensis* dominance,
before *A. skvortzowii* once again asserts its dominance by the end of the Younger Dryas,
alongside notable increases in *C. inconspicua* and *L. minuta*, resulting in increasing richness
($N0$) and diversity ($N2$) scores (Fig. 4). These changes are reflected in a rapid change in DCA
axis 1 scores at the start of the Younger Dryas, with discernible changes in the evenness ratio
indicative of increasing importance of different species. The Early Holocene is characterised
in the main by declining dominance of *A. skvortzowii* and increasing importance of autumn
blooming crops of *C. inconspicua* and *Lindavia minuta*. *U. acus* shows considerable
fluctuations, with *Hannaea baicalensis* and *Cyclostephanos dubius* increasing in importance
from c. 10 cal kyr BP, driving increases in diatom $N2$ diversity and evenness ratio.

Where the climatic periods are used as a supervised classification technique, MCT analysis
reveals the highest variation between the Early and the Middle Holocene subepochs (Fig. 5).
This transition also marks the time when the evenness ratio peaks, that then declines to
lowest values found in our study during the Middle and Late Holocene (Fig. 4). During the
Middle Holocene, several diatom species largely disappear from the stratigraphical record
including *Stephanodiscus flabellatus*, *Stephanodiscus skabitchevskii*, *H. baicalensis*, and *C.*
*dubius*, while *C. inconspicua* drops to low values for the remainder of the record (Fig. 3). In
their place, *U. acus* and *L. minuta* increase in importance. The loss and gain of species
through the Middle Holocene has resulted in a relatively high but stable $N2$ diversity, with
peak diversity occurring at the boundary between the Middle and Late Holocene, c. 4.2 cal
kyr BP (Fig. 4). Despite relatively high abundances for the first time of *Stephanodiscus*
*parvus* and *Stephanodiscus meyerii*, the Late Holocene sees *L. minuta* increasing to almost
complete dominance of the diatom assemblage between 1.8–1.4 cal kyr BP, resulting in some
of the lowest diatom diversity values for the study. During the most recent 500 years of our
record, the assemblage is characterised by species commonly found in the modern–day lake
communities, co-dominated by *A. baicalensis*, *U. acus* and *L. minuta* (Fig. 3).

Palaeoproductivity
Diatom biovolume accumulate rate (BVAR) data show that spring diatom crops have
dominated palaeoproductivity for much of the past c. 16,000 years, apart from a short period
during the late Holocene, 1.8–1.4 cal kyr BP, when autumnal diatom production increases to
more than ten times that of spring production (Fig. 6). BVAR were substantially higher and
more unstable during the Younger Dryas stadial than the warm periods before and after it
(Fig. 6). Peak palaeoproductivity for the whole study ($18.3 \times 10^3$ $\mu m^3$ $cm^{-2}$ $yr^{-1}$) was
associated with successive peaks in *U. acus* then *A. baicalensis* at c. 12.3 cal kyr BP. BVAR
then declined rapidly and remained relatively low for the remainder of the Younger Dryas
and the start of the Holocene. Palaeoproductivity increased during the Early and Middle
Holocene, before declining to lowest rates during the Late Holocene (mean, $1.3 \times 10^3$ $\mu m^3$
$cm^{-2}$ $yr^{-1}$).

The shapes of the productivity–diversity relationships differ among the pre-defined climate
periods, with some being negative (e.g. the Younger Dryas and the Middle Holocene
periods), positive (e.g. the pre-BA and the Early Holocene periods), and others more complex
such as the Late Holocene, dependent on palaeoproductivity rates (Fig. 7). The relationship
between palaeoproductivity and diversity during the Bølling–Allerød was flat (Smith 2007).
GAMs with very low *p*-values suggest that we have higher confidence that PDR relationships
existed during the Younger Dryas, the Early Holocene, and the Late Holocene in comparison
to the other time periods (Table 1).


## 4. Discussion

Over the past 16,000 years diatom trends in Lake Baikal have been dynamic, with decadal shifts in species dominance punctuating millennial-scale trends in seasonal communities. MCT analyses show that the main split in diatom composition occurs between assemblages associated with the Last Glacial – Interglacial Transition (LGIT; 16–8.2 cal kyr BP), and the Middle– Late– Holocene subepochs (8.2 cal kyr–present) (Fig. 5). This suggests that trends in community shifts in diatoms in Lake Baikal broadly reflect climatic changes associated with the demise of Northern Hemisphere ice sheets as the planet transgressed from glacial to interglacial environments (Denton et al. 2010). Here we focus on interpreting trends in diatom communities and diversity through time, before focussing on how productivity–diversity relationships (PDR) vary over long-timescales, including during periods of rapid climate change and periods of relative climate stability.

4.1 Last Glacial – Interglacial Transition

Due to high obliquity (Fig. 8j), the LGIT in the Lake Baikal region was intensely seasonal, resulting in very cold winters but relatively warm, moist summers (Bush, 2005). Cold winters during the late glacial resulted in substantially longer annual ice cover on Lake Baikal of up to 8–9 months, reduced hydrological input by as much as 40 % than the modern day, and restricted inputs of key nutrients such as P, N and Si (Shimaraev & Mizandrontsev 2006). Up to the Bølling–Allerød, diatom assemblages were dominated by spores of the endemic *Aulacoseira skvortzowii* (Fig. 3). *A. skvortzowii* is tychoplanktonic, having evolved a life cycle which utilizes both pelagic and littoral habitats; viable spores remain in littoral sediments down to a depth of 25 m, where they can be resuspended by strong autumn storms back into the pelagic zone (Jewson et al., 2008) in time to bloom the following spring. *A. skvortzowii* grows best in water temperatures below 5 °C, and to avoid lethal increasing surface water temperatures, spore formation is triggered after ice break-up when phosphate concentrations fall below a threshold level of 15–23 µg L$^{-1}$ (Jewson et al., 2008). Declining phosphate concentrations are caused by P uptake by other algae (e.g. green algae and picoplankton) as surface waters start of stratify. Pigment analyses have shown that green algae were abundant at this time in the south basin of Lake Baikal (Tani et al. 2002; Soma et al. 2007). Relatively low $N2$ plankton diversity and evenness values during much of the late

glacial likely reflect P and Si limitation, resulting from overall lower nutrient concentrations due to colder climate (Shimaraev & Mizandrontsev 2006) but possibly also competition from other algae.

The major shift to warmer temperatures during the LGIT occurred with the onset of the Bølling–Allerød interstadial, c. 14.7 cal kyr BP, linked to heat release from warm waters deep in the North Atlantic ocean (Thiagarajan et al. 2014) which led to the resumption of the Atlantic meridional overturning circulation (AMOC) (McManus et al. 2004) (Fig. 8g). Mean pollen-inferred temperatures of the warmest month from neighbouring Lake Kotokel however (Fig. 8b), reveal a 5 °C increase only during the latter stages of the Bølling–Allerød (Tarasov et al. 2009). At the same time, pollen-inferred mean annual precipitation records show that rainfall increased from c. 300 mm $yr^{-1}$ to c. 460 mm $yr^{-1}$ (Fig. 8c). This warmer, wetter Allerød led to extensive melting of glaciers along the shores of Lake Baikal (Horiuchi et al. 2004; Osipov and Khlystov 2010), major expansion of boreal vegetation (Tarasov et al. 2009; Bezrukova et al. 2010; Reshetova et al. 2013), reduced ice duration on the lake and increased nutrient transport (Shimaraev & Mizandrontsev 2006). The impacts on diatom communities however were subtle, with diatom composition between the pre– Bølling– Allerød and Bølling–Allerød not distinguishable through MCT analyses (Fig. 5). Diversity and evenness show little variability, although relative abundances of a few species show distinct changes. For example, the heavily silicified endemic *Aulacoseira baicalensis* will have benefited from increased Si availability (Jewson et al. 2010) while the endemic *Stephanodiscus flabellatus* may have benefited from elevated concentrations of phosphorus (Bradbury et al. 1994). The observed decline in *A. skvortzowii* spore formation may also be attributable to general increased P availability in the south basin (Jewson et al. 2008).

The Younger Dryas stadial (12.9 – 11.7 cal kyr BP), is probably the most studied climate disturbance event of the late Quaternary (Broecker et al. 2010). It was caused by an influx of freshwater from Lake Agassiz into the Arctic Ocean (Tarasov & Peltier 2005; Murton et al. 2010) which led to the decline in AMOC (Fig. 8g) (Bond et al. 2001; McManus et al. 2004), and concomitant cooler temperatures across the Northern Hemisphere (Stuiver et al. 1995) (Fig. 8h). Increased GISP2 $K^+$ concentrations (Fig. 8f) indicate that the Siberian High intensified at the start of the Younger Dryas (Mayewski et al. 1997), which accounts for both an attenuated Asian summer monsoon (Fig. 8e) (Dykoski et al. 2005) and cooler and more

arid climate conditions in the Lake Baikal region (Tarasov et al. 2009) (Fig. 8 b, c). These
cooler, more arid conditions led to a decline in hydrological input into Lake Baikal itself
Mackay et al. 2011), and a short-lived expansion of the tundra biome (Tarasov et al. 2009;
Bezrukova et al. 2010).

Diatom responses to climate change within the Younger Dryas were instantaneous but
complex, kick-started by the first appearance and rapid growth of *Ulnaria acus* (Fig. 3)
(indicative of higher dissolved silica concentrations in the water column (Bradbury et al.
1994)), and increasing abundance of *C. inconspicua*. However, the decline in $N2$ diversity
only from c. 12.6 cal kyr BP (Fig. 4), suggests that resources for diatom growth were not
initially limiting. Occupying different seasonal niches, spring blooming *U. acus* and
autumnal blooming *C. inconspicua* (Ryves et al. 2003) are unlikely to be directly competing
for the same resources, in part because small centric diatoms do not utilise a lot of silica
(Bradbury et al. 1994). Having several co-dominant species is reflected in the relatively high
evenness scores for the Younger Dryas (Fig. 4), related to few resources being limiting
(Interlandi & Kilham 2001).

The decline in $N2$ diversity is linked to peak values for *A. baicalensis* (Fig. 3), competing
directly with *U. acus* for dissolved silica. Diatom population changes in Lake Baikal play an
important role in controlling availability of silicic acid for other species (Callender & Granina
1995; Jewson et al., 2010; Shimaraev and Domysheva, 2013; Jewson et al., 2015). In
particular, when populations of *A. baicalensis* are very high, the availability of silicic acid
uptake for other species declines, leading to a drop in their populations (Jewson et al., 2010;
2015). Being a heavily silicified species, *A. baicalensis* needs a strong period of mixing
(Huisman et al. 2004), which suggests that when abundant, turnover is strong, allowing the
recycling of nutrients needed for peak palaeoproductivity (Fig. 6). Our diatom records
demonstrate that species specially adapted to grow under the ice of Lake Baikal (Bondarenko
et al. 2012) have flourished alongside autumnal blooming species during this cold stadial,
with, initially at least, few limiting resources.

The latter stages of the Younger Dryas are characterised by a marked increase in diatom
richness and diversity associated with both increases in spring and autumn diatoms, and a
rapid decline in palaeoproductivity. Given that regional temperatures remain low and
unchanged at this time (Demske et al. 2009), internal dynamics in the lake may be driving
observed changes in diatom communities, mediating resource availability. Contemporary
monitoring studies have shown that populations of *A. baicalensis* can undergo dramatic
changes in population size linked to complex interactions including silica availability
(Izmest'eva et al. 2006); after populations peak, cells are transported to the bottom sediments,
taking with them silica that then becomes unavailable for new diatom growth (Jewson and
Granin 2015). But autumnal diatoms are still able to grow, as nutrients including Si become
available once more during autumnal turnover.

As detailed above, MCT analyses is unable to distinguish Younger Dryas and Early Holocene
diatom communities (Fig. 5), two time periods characterised by significant millennial scale
variability (Bond et al., 2001; Fisher *et al.*, 2002; Mayewski et al. 2004; Nesje et al. 2005;
Wanner et al. 2014). Overall, the diatom flora is characterised by persistent spring-blooming
species that need lots of Si, but also increasing abundance of autumnal blooming diatoms. A
decline in species richness and diversity at the start of the Early Holocene may be indicative
of fewer resources (Interlandi & Kilham 2001), coincident at least with the widely expressed
cool Preboreal Oscillation (Fisher et al. 2002) and lower river flow into Baikal's south basin
(Mackay et al. 2011). After c. 11.3 cal kyr BP diatom diversity and evenness increase for the
next few thousand years, indicative of increasingly abundant resources, linked to regional
warmer and wetter climates (Tarasov et al. 2009), and generally stronger seasonality. Co–
limitation by several nutrients and light allow for greater numbers of planktonic species to
co-exist (Burson et al. 2018), leading to highest evenness scores at the Early / Middle
Holocene boundary.

4.2 Middle – Late Holocene

Northern Hemisphere cooling, especially from c. 5 cal kyr BP, is linked to declining summer
insolation (Marcott et al. 2013) in conjunction with vegetation and snow/ice feedbacks on
Earth's surface albedo. This cooling culminated in the neoglacial, which in central Asia
resulted in increased aridity (Ganopolski et al. 1998) expressed in the Lake Baikal region as
declining mean annual precipitation (Tarasov et al. 2009). The neoglacial also resulted in
increased Siberian permafrost formation (Anisimov et al. 2002). Within Lake Baikal, we
document a shift from spring to autumnal diatom communities that started during the Early
Holocene, concomitant with the first decline in insolation, but then continues to get stronger
during the Middle and Late Holocene as both precession and obliquity decline (Fig. 8). The
slow unfolding of an increasingly important autumnal diatom community has relatively little
impact on palaeoproductivity during much of the Holocene, although autumnal production
after 10 cal kyr BP is slightly higher than it was before 10 cal kyr BP, and continues to
proportionally increase especially during the Late Holocene. This long-term change in
diversity and palaeoproductivity likely represents an orbitally-driven ecosystem response to
long-term climate change, possibly linked to changes in seasonality and its impact on, for
example, ice cover dynamics and length of summer stratification on the lake.

*L. minuta* is currently the only pelagic diatom to bloom during the autumn in Lake Baikal in
substantial numbers (although it can also grow in smaller numbers during spring turnover).
Because *L. minuta* can tolerate surface water temperatures up to 11 °C, cells persist higher in
the surface waters for longer during summer stratification, so that when stratification breaks
down in the autumn and nutrients are regenerated, cells of *L. minuta* are retrained into the
photic zone first, giving them a strong competitive advantage, leading to their dominance
during autumn turnover (Jewson et al., 2015). Winter ice cover is a major force driving inter-
seasonal connections in lakes that freeze over every year (Sommer et al. 2012; O'Reilly et al.
2015; Hampton et al. 2017), and Lake Baikal is no exception. The gradual decline in
proportion of spring diatoms may have resulted in increased resource availability for other
diatoms (Interlandi & Kilham 2001; Jewson et al., 2010; 2015). For example, monitoring
studies have shown large autumnal populations of *L. minuta* during the 1950s CE when
spring populations of *A. baicalensis* were very low (Antipova & Kozhov in Kozhov 1963).

The build-up to peak dominance in *L. minuta* during the Late Holocene is accompanied by
the successive decline in spring blooming species, starting with *U. acus* followed by *A.*
*baicalensis*, species that both rely on availability of dissolved silica (Jewson et al. 2010) and
under-ice turbulence to remain in the photic zone (Granin et al. 2000). These shifts are
reflected in rapidly declining $N2$ diversity scores since their peak at the Middle / Late
Holocene boundary to their lowest values for the Holocene (Fig. 4). This decline is also
coincident with a major shift in carbon dynamics in Lake Baikal, linked to increasing
regional cooling and aridity (Mackay et al. 2017). The almost monospecific Late Holocene
assemblage of *L. minuta* and resultant low $N2$ diatom diversity in the lake is concurrent with
the Dark Ages Cold Period (DCAP) (1.8–1.4 cal kyr BP; CE 400–765) (Helama et al. 2017),
a climatically cool period, linked to changes in ocean circulation (Bond et al. 2001), and solar
and volcanic activity (Helama et al. 2017). The near complete disappearance of *A.*
*baicalensis* makes this period unique in our Lake Baikal record. However, we are not yet able
to say conclusively why growth of *A. baicalensis* was so inhibited during the DCAP, leading
to the dominance of *L. minuta* for several centuries. This period almost certainly represents
an ecosystem response to abrupt extrinsic change occurring elsewhere in the northern
hemisphere, super-imposed on longer-term changes in orbital parameters (Williams et al.
2011). It is possible that the smaller biovolume of *L. minuta* in comparison to *A. baicalensis*
and *U. acus*, may have conferred it a competitive advantage as resources became increasingly
limited (Burson et al. 2018). Alternatively, persistent deeper snow cover on the frozen lake
could have acted to reduce light penetration through the ice, (i.e. light as a resource is
severely limited) resulting in a decline in sub-surface turbulence and loss of spring diatom
crops to deeper waters (Granin et al., 2000). It is possible that these responses are linked to
orbitally driven seasonal changes such as cooler summers (Fig. 8). It's notable that the
ecosystem function of primary production did not decline overall during this period, just the
timing of peak production, with autumnal palaeoproductivity far exceeding spring
productivity for the only time in our record during the DCAP (Fig. 6).
4.3 Palaeoproductivity–diversity relationships
Productivity–diversity relationships (PDR) in aquatic ecosystems are complex (Aoki 2003;
Smith 2007), while also being scale (Chase and Leibold 2002; Korhonen et al. 2011) and
time (Rusak et al. 2004) dependent. Our initial hypotheses set out to test if these relationships
differ between periods of relative climate stability and periods of rapid climate change. This
is borne out, as the shapes of the relationships are very different across the different climate
periods (Fig. 7). The strongest evidence for GAMs (with lowest $p$ values; (Table 1)) were
found during both the Younger Dryas and Early Holocene, periods punctuated by pervasive
millennial-scale variability, and the Late Holocene period, characterised by cold events
linked to overlapping drivers such as solar minima and volcanic eruptions (Helama et al.
2017), amplified by centennial-scale oceanic variability (Renssen et al. 2006). PDR were
different during each of the pre– Bølling–Allerød, the Bølling–Allerød and the Middle
Holocene periods, but statistical support for GAMs were much weaker during these periods
of relative climate stability (Table 1). During the Bølling–Allerød the PDR is effectively flat
(sensu Smith 2007), because $N2$ diversity is restricted to very low values and changes little
across the magnitude of the productivity gradient, perhaps due to as yet few available
resources for several species to co-exist. During the Middle Holocene, $N2$ diversity values are
again more restricted than during either the Early or Late Holocene periods, $N2$ are rarely low
(Fig. 7), likely linked to optimal resource availability allowing several species to co–exist
(Burson et al. 2018) at similar abundances (Fig. 4) during both spring and autumn (Fig. 3).

The negative PDR during the Younger Dryas likely reflects a complex set of processes linked
to both ecosystem responses to abrupt cooling (e.g. increased ice duration but stronger
overturn in the lake (Shimaraev & Mizandrontsev 2006), and a reduction in resource
heterogeneity due to severe limitation by a few or even single resource (Interlandi & Kilham
2001). Productivity in Lake Baikal is dependent on a sustained supply of nutrients from
deeper waters to the photic zone, stimulated by shifting wind dynamics and enhanced deep
ventilation (Swann et al. 2020). While increased supply of Si ensured initial resources for
*Ulnaria* and *Aulacoseira* growth, Si will ultimately have become unavailable to most other
diatoms, due to massive increases in the abundances of *A. baicalensis* (Fig. 3) (Jewson et al.,
2010; 2015), leading to declines in diversity while productivity peaked. During the Early
Holocene however, there is a positive PDR, especially when palaeoproductivity levels are
initially lower. We suggest that rapid melting of glaciers and increased fluvial input (Osipov
and Khlystov 2010), and opening up of new habitats as ice-cover duration on the lake
declined, contributed to an increased supply of nutrients and habitat availability, such that
both diversity and productivity were able to increase from the very low values at the start of
the Holocene period. It's notable that during the Early Holocene the ratio of autumnal to
spring species increases markedly, indicative of autumnal overturn niches opening up,
allowing increased growth of *C. inconspicua* and *L. minuta* (Fig. 3).

During the Late Holocene however, a negative PDR is apparent only when
palaeoproductivity is very low at the transition between the Middle and Late Holocene
periods  (Fig. 7). Low palaeoproductivity at c. 4.2 cal kyr may be linked to a period of
regional cooling and aridity (Mackay et al. 2017) concurrent with weaker Asian summer
monsoons (Dixit et al. 2014). It is at this time that new *Stephanodiscus* species become
established in the lake (*S. meyeri* and *S. parvus*) alongside existing diatom communities,
hence higher $N2$ diversity. *Stephanodiscus* species tend to reflect higher phosphorus than
silicon loading rates (Kilham et al. 1986), and may be a community response to cooler, more
arid climate (Bradbury et al. 1994) as experienced during the Late Holocene neoglacial.
However for the majority of the Late Holocene there is little relationship between

625 palaeoproductivity and diversity, indicative of abundant resources allowing for many

626 planktonic species to co–exist.


628 4.4 Comparisons to other studies within Lake Baikal


630 It has long been recognised that Quaternary biogenic silica and diatom concentrations in Lake

631 Baikal sediments mirror changes in insolation (Khursevich et al. 2001), such that very low

632 concentrations characterise glacial periods, likely due to a number of factors including lower

633 productivity and higher rates of diatom dissolution, as well as dilution due to increased inputs

634 of clastic material (Mackay 2007). The Vydrino LGIT record has an almost identical diatom

635 assemblage to those identified for the same time period in long cores extracted from

636 elsewhere in the lake, including both the Posolskaya Bank (BDP-99) and Academician Ridge

637 (BDP–96–2) (Khursevich et al. 2005) (Fig. 1). In another study of Quaternary Lake Baikal

638 diatoms, this time from the Buguldieka Saddle (Fig. 1), Edlund (2006) found that although

639 earlier glaciations also contained few diatoms, the 'Sartan glaciation', i.e. Marine Isotope

640 Stage 2, still contained at least 10 species of planktonic diatoms, and an assemblage again

641 very similar to our Vydrino sequence. Bradbury et al. (1994) produced a similar but much

642 lower resolution record for diatom changes spanning the past 15,000 years from station 305

643 off the southern margins of the Selenga Delta (Fig. 1), where both the assemblage and

644 sequence of diatoms are similar to Vydrino. However, while observations and conclusions

645 drawn in this study related to PDR are likely applicable to other regions of this vast lake, one

646 of the reasons why Lake Baikal contains a diverse endemic diatom flora is down to its size

647 and heterogeneity of habitats (Jewson et al. 2015). Thus while species declines may be

648 evident in one region, those same species may well persist in other regions of the lake.

649

650 Conclusions

651

652 This study provides important insights into our understanding of productivity–diversity

653 relationships (PDR) in aquatic ecosystems. We show that diatom communities and

654 palaeoproductivity are sensitive to extrinsic drivers of climate change. These drivers operate

655 at different time scales, from abrupt events during the Younger Dryas resulting in (negative)

656 coupling between palaeoproductivity and diversity, to slower changes in boreal insolation

657 and seasonality, leading to varied PDR relationships. These differences are likely related to

658 resource availability and limitation (or not) of specific nutrients, linked to climate and surface

water overturn. We therefore provide important perspective to complement existing short-
term observational (Dodson et al. 2000) and experimental (Winfree et al. 2015) approaches.
PDR in aquatic systems should not be expected to occur in one direction (Smith 2007), but
are very much dependent on other factors such as external disturbances, resource availability,
species interactions and abiotic constraints on ecosystem function. Even given the sheer
volume of Lake Baikal, diatom responses to abrupt events can be almost instantaneous,
showing how tightly coupled ecology and climate have been in the past.

Over sub-orbital timescales, our records suggest that ecosystem function in Lake Baikal is
rather resilient to natural disturbance. Following the concept of operation criteria as defined
by Jovanovska et al. (2016), after disturbance or "press" events like the Younger Dryas,
diatom communities and palaeoproductivity return to pre-disturbance states. Moreover, rather
than leading to the demise of certain species, new species are actually ushered into the palaeo
record, likely due to increased resources (Fig. 3). And even when a regime shift looks
possible with respect to the increasing dominance of an autumnal flora as the Holocene
unfolded, in the past 1000 years a spring-dominated flora has returned. These observations
may hold insights into observed diversity changes occurring in recent decades, linked to
global warming (Roberts et al., 2018). However, what our record cannot provide information
on is the resilience of the lake's flora to multiple stressors such as human driven climate
change and anthropogenic nutrient enrichment, which is currently impacting the lake's
littoral communities.

**Author Contributions:**
Study was conceived by AWM and PR. Diatom counting was undertaken by DWM. Age
modelling was done by NP. Data analyses was undertaken by VAF, AWRS and AWM.
AWM led the paper writing with significant contributions from AWRS and GEAS. All
authors made comments on earlier drafts of the paper.


**Acknowledgements:**
We wish to acknowledge the various agencies who helped to fund this work, especially the
EU FPV programme (EVK2-CT-2000-0057), UK NERC (NE/J010227/1), and the
Norwegian Research Council (IGNEX ref: 249894/F20). We thank UCL Geography
Cartography Unit who helped prepare the figures. And we especially thank the three
reviewers whose observations and comments have helped to improve the focus of the study
immensely. AWM would like to say a special thanks to the editorial team of CP guiding the
study's publication over an extended period of time due to illness. The flexibility and
understanding shown should be a model for other journals to follow.

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

Table 1: GAMs to investigate PDR trends during pre-defined climate periods. edf = effective
degrees of freedom, and ref.df = reference degrees of freedom. The goodness-of-fit statistics
show the adjusted $r^2 = 0.462$ and the deviance explained is 53.2 %.

| Climate period | edf | ref.df | F | *p*–value |
|---|---|---|---|---|
| **Late Holocene** | 3.719 | 4.481 | 5.242 | 0.0002 *** |
| **Middle Holocene** | 1.001 | 1.002 | 6.672 | 0.0100 * |
| **Early Holocene** | 2.291 | 2.916 | 7.183 | 0.0001 *** |
| **Younger Dryas** | 1.001 | 1.001 | 20.168 | 9.18e-06 *** |
| **Bølling–Allerød** | 1.105 | 1.203 | 0.652 | 0.4030 |
| **pre-BA** | 2.562 | 3.167 | 2.760 | 0.0443 * |

Significance codes:  0 '***' 0.001 '**' 0.01 '*' 0.05 '.' 0.1 ' ' 1



**Figure Legends:**

Fig. 1: Map of Lake Baikal and its catchment, with locations of the different cores used or mentioned
in this paper, including the coring location on the Vydrino Shoulder, where our study was undertaken.
Also shown is Lake Kotokel that has provided independent climate reconstructions used in this study
(Tarasov et al. 2009).

Fig. 2: 'Bacon' Age-depth model (Blaauw & Christen, 2011) for Vydrino box (CON01–605–05) and
piston cores (CON01–605–03) of radiocarbon AMS dates calibrated using IntCal20 radiocarbon
calibration curve (Reimer et al., 2020).

Fig. 3: Relative abundances of sub-fossil planktonic and tychoplanktonic diatoms spanning the past c.
16,000 years from Vydrino Shoulder. Diatoms which grow mainly before summer stratification are
shown in blue, while those that mainly grow after summer stratification are shown in green. DCA axis
1 sample scores summarize the major trend in species turnover.

Fig. 4: Comparison of Hill's richness ($N$0), diversity ($N$1 and $N$2) and evenness ($N$2/$N$1 ratio) trends
over the past c. 16,000 years. Richness and diversity values were further divided by sediment
accumulation rates over time, so these measures become estimated Hill's numbers per $cm^{-2}$ $yr^{-1}$

Fig. 5: Multivariate classification tree (MCT) for Lake Baikal planktonic and tychoplanktonic diatoms
over the past c. 16,000 years. Time periods used in multivariate classification tree analyses are: the
pre Bølling–Allerød (pre–BA), the Bølling–Allerød (B–A) interstadial, the Younger Dryas stadial and
the three recently ratified Early, Middle and Late Holocene subepochs (Walker et al. 2018). The
histograms show the transformed abundances of the discriminating taxa within each leaf. These are
listed in decreasing order. The numbers under the histograms represent the relative error within each
leaf, and $n =$ the total number of samples. At the bottom is the summary of the residual error (RE),
cross-validated error (CVRE), and standard error (SE). The $r^2$ will be 1-RE.

Fig. 6: Stratigraphical profile showing trends in palaeoproductivity over the past c. 16,000 years: the
proportion of diatoms that bloom in the autumn versus those that bloom in the spring; estimates of
palaeoproductivity derived from spring (orange) and autumn (dark brown) diatom biovolume
accumulate rates (BVAR $\mu m^3$ $cm^{-2}$ $yr^{-1}$); and log-transformed ratio of autumnal / spring
palaeoproductivity.

Fig. 7: Palaeoproductivity–diversity relationships explored using generalized additive models
(GAMs) that allow different smooths for the pre–defined climatic periods: pre Bølling–Allerød,
(Bølling–Allerød), Younger Dryas, Early Holocene, Middle Holocene, and Late Holocene. The dots
represent the data points, the thick line is the fitted GAM response for each period, and the shaded
areas represent the 95 % confidence intervals around the mean fitted response.

Fig. 8: Multi-archive data plotted alongside (a) DCA axis 1 sample scores of Lake Baikal diatoms, as
a measure of turnover over the past c. 16,000 years; (b) mean pollen–inferred annual precipitation
from Lake Kotokel, with smooth line representing the mean 3–point moving average (Tarasov et al.
2009); (c) mean pollen–inferred temperature of the warmest month from Lake Kotokel, with smooth
line representing the mean 3–point moving average (Tarasov et al. 2009); (d) mean Northern
Hemisphere temperature stack records for 60° latitude bands (30° N – 90° N; Marcott et al. 2013); (e)
$\delta^{18}O$ values of Dongge Cave stalagmite D4 (Dykoski et al. 2005); (f) $K^+$ ion concentrations (ppb)
from GISP2 D core (Mayewski et al. 1997); (g) North Atlantic core GGC5 $^{231}Pa/^{230}Th$ meridional
circulation data; (h) $\delta^{18}O$ values of NGRIP ice core (Rasmussen et al. 2006); (i) June insolation 60° N
(W m$^{-2}$; Berger & Loutre, 1991); (j) obliquity ($\varepsilon$).

Fig. 1

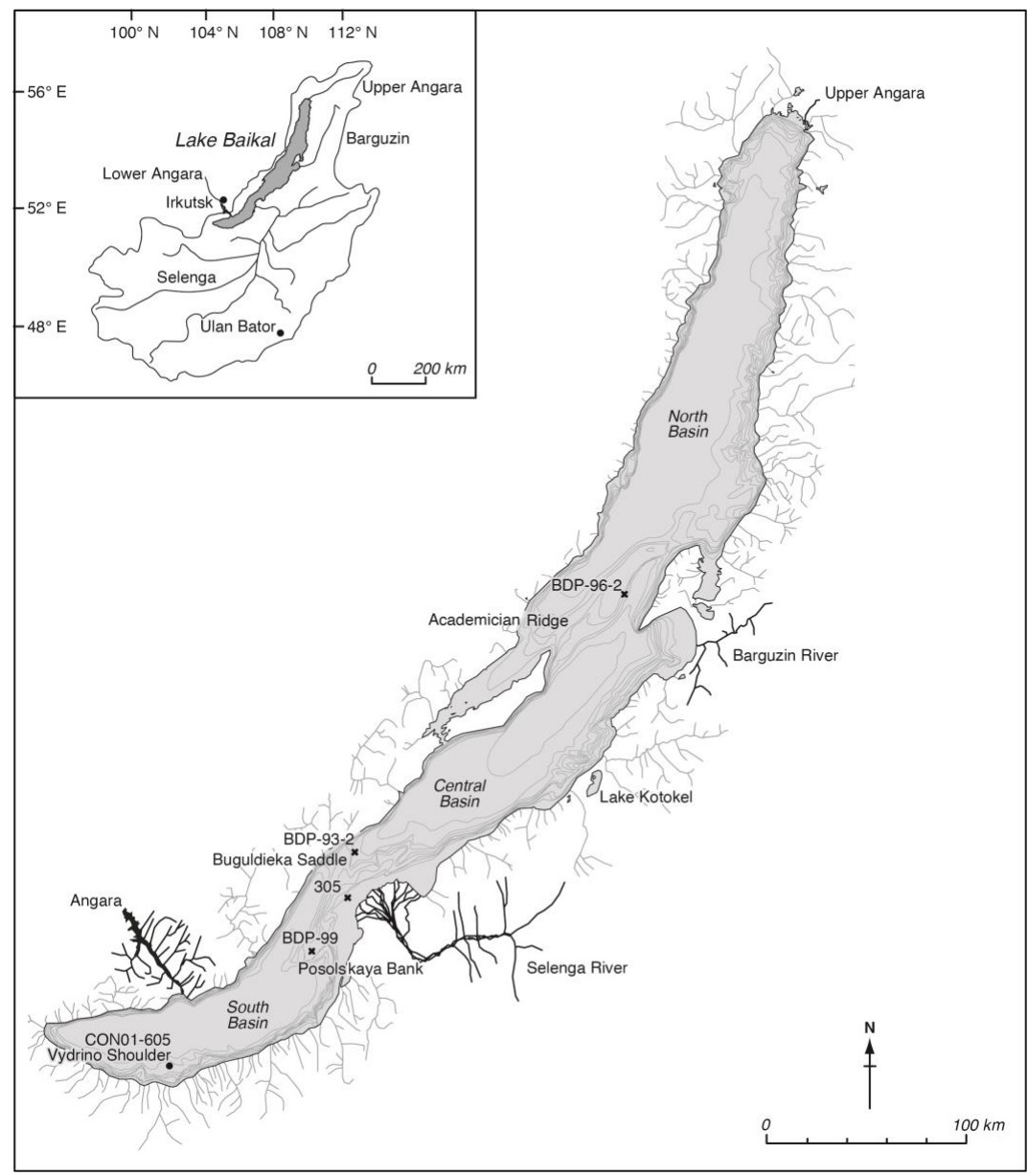


Fig. 2

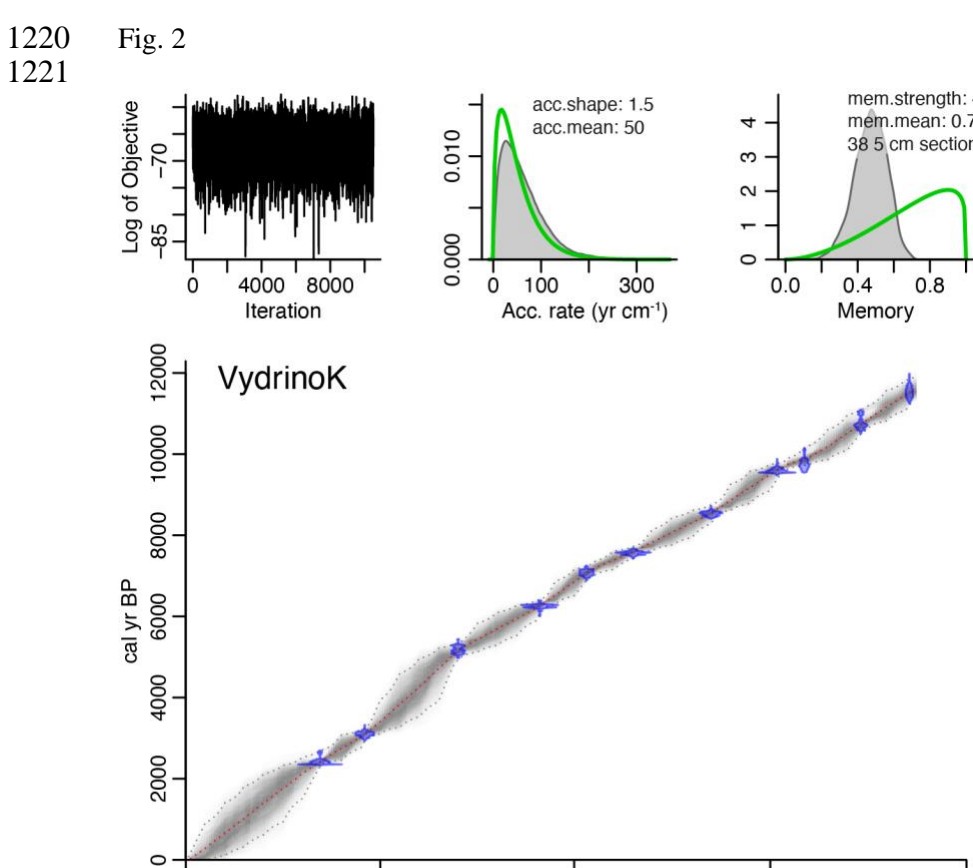


Fig. 3

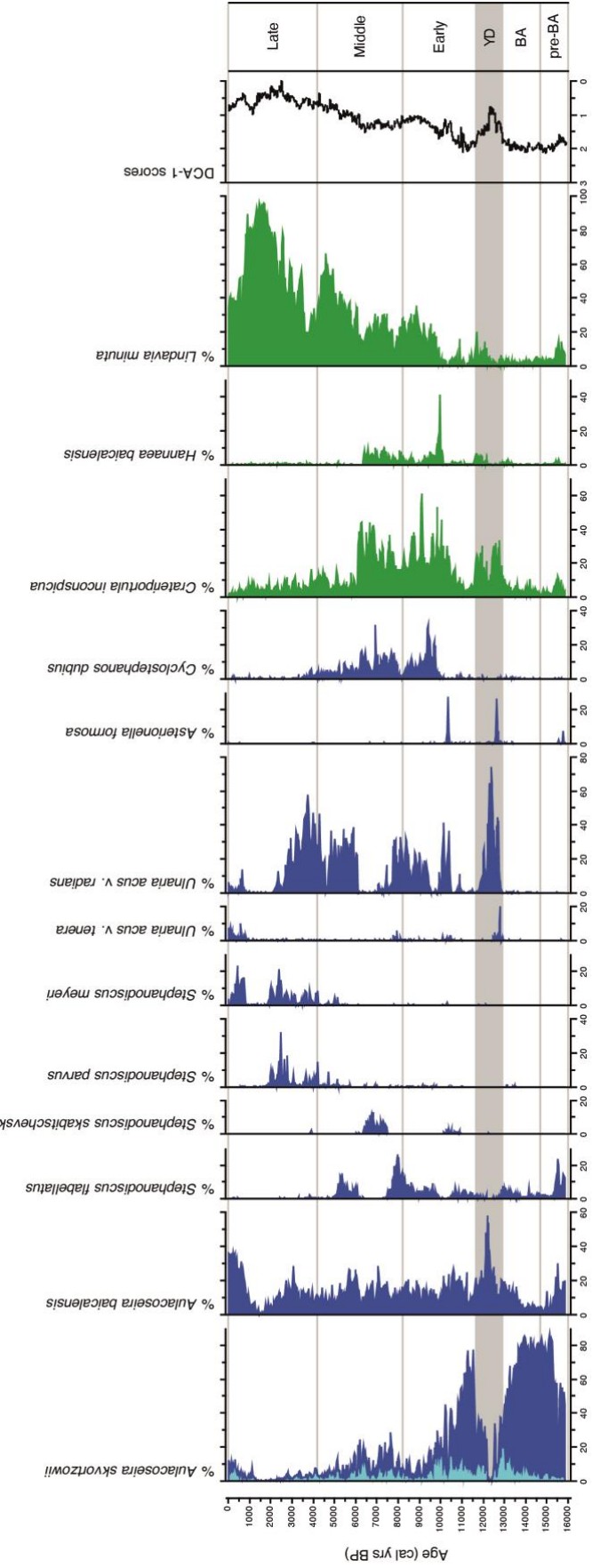


Fig. 4

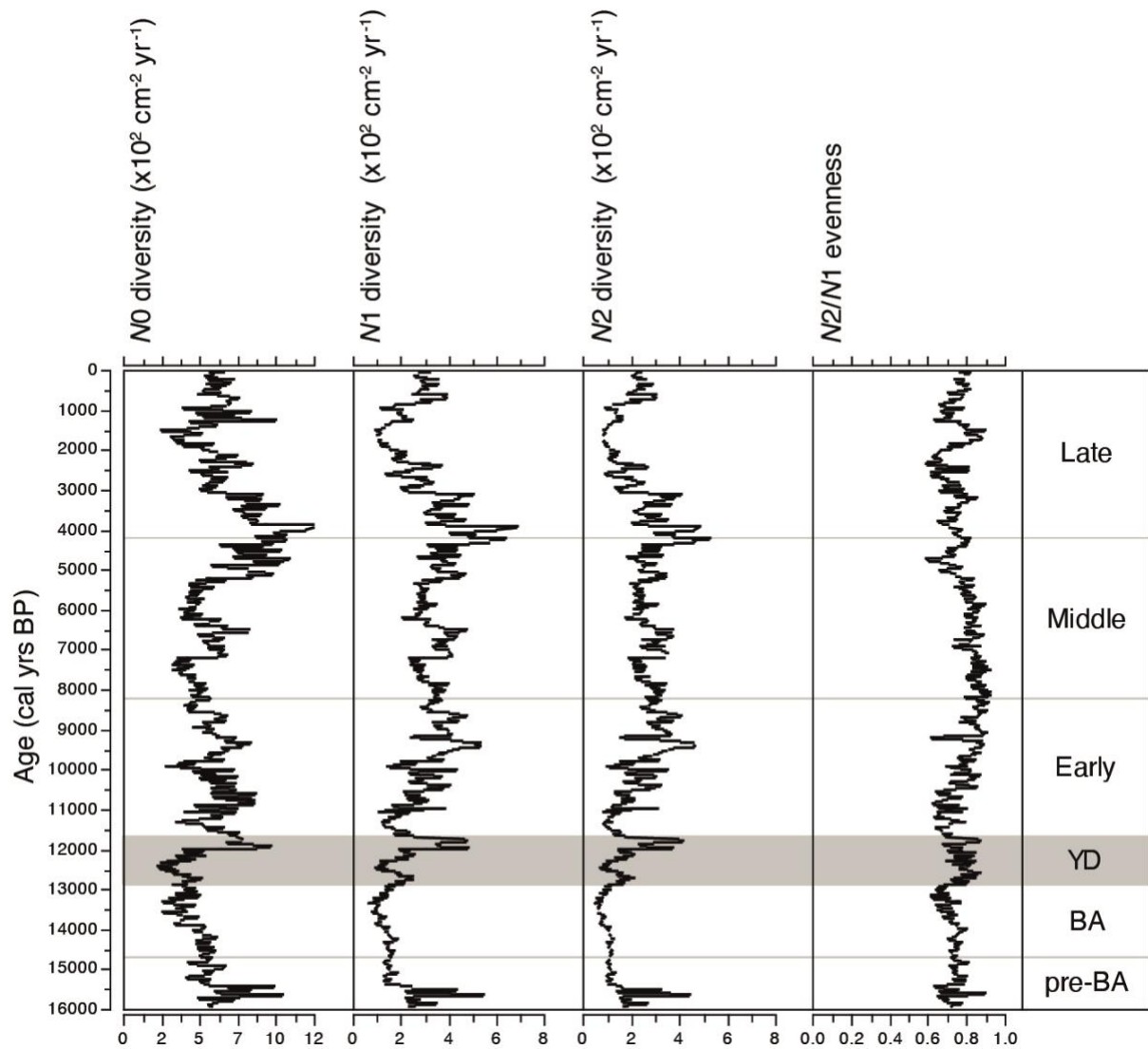


Fig. 5

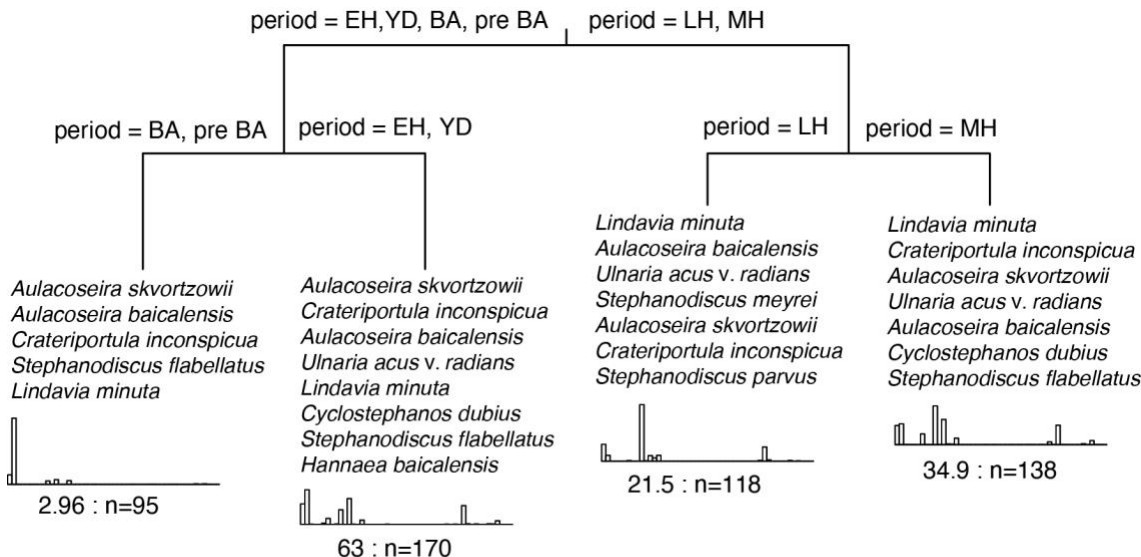


Fig. 6

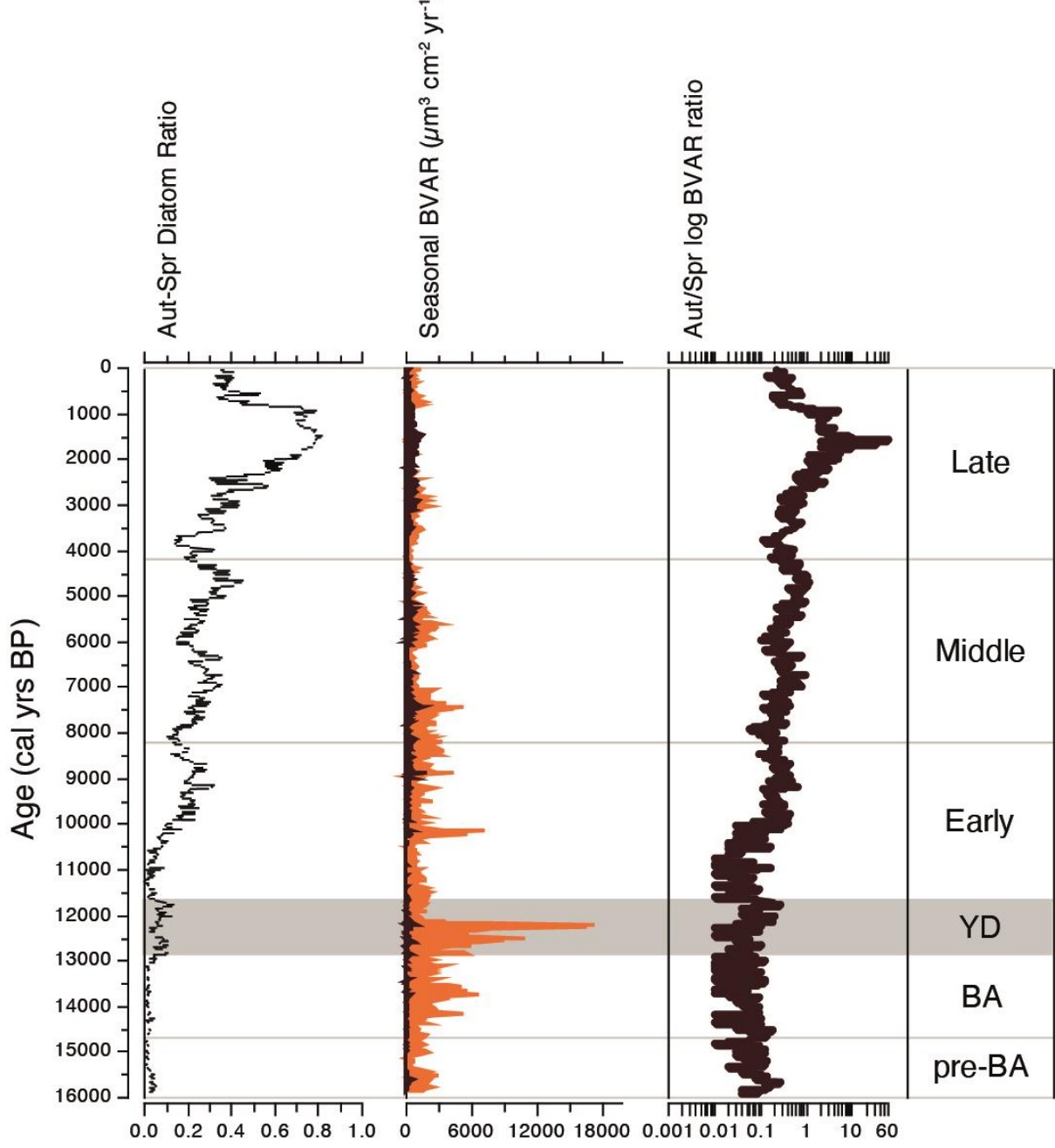


Fig. 7

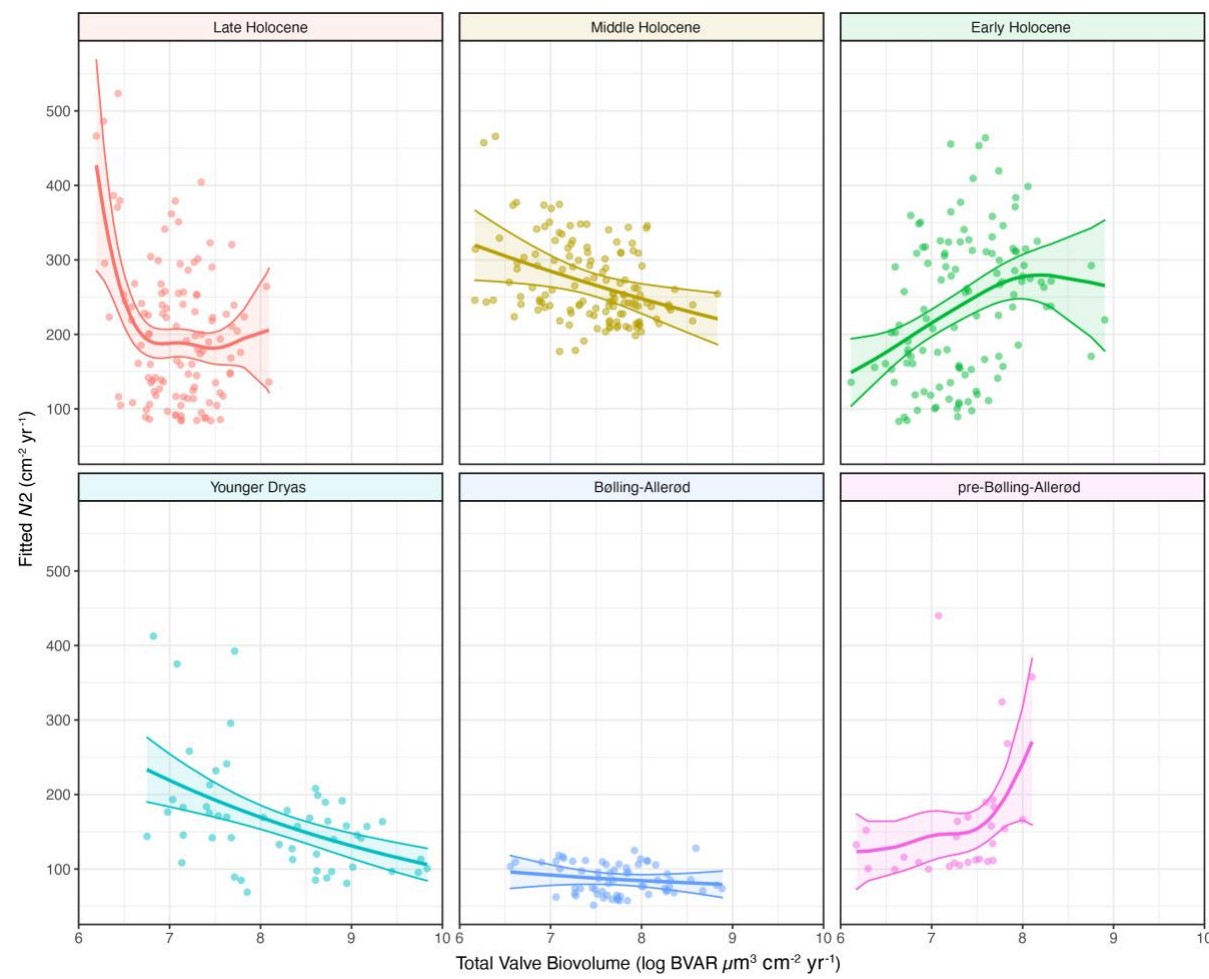


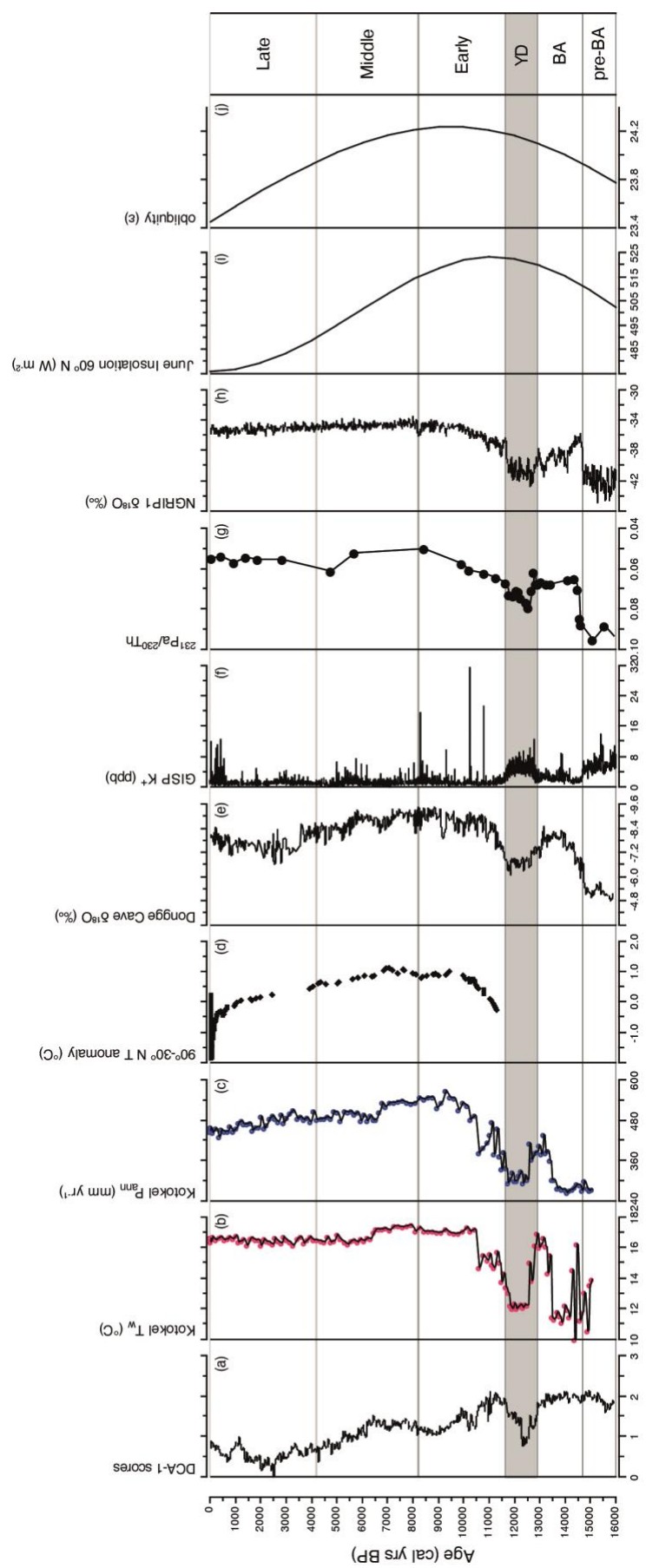