# Peer review of "Long term trends in diatom diversity and palaeoproductivity: a 16,000-1 vear multidecadal study from Lake Baikal, southern Siberia 2 3 4 5 6 Anson W. Mackay1\*, Vivian A. Felde2, David W. Morley1, Natalia Piotrowska3, Patrick 7 Rioual4, Alistair"

_Climate of the Past, 2020_

## Short Comment (SC1) · 30 Jun 2020

Mackay and colleagues present data on diatom abundances and associated data from Lake Baikal, with a focus on long term trends in the diversity, productivity, and stability of the diatom record.

I have a couple of technical comments to make regarding the methods used, which speak to the appropriateness of the methods and therefore the evidence they provide in support of the authors interpretations.

[Figure]

My first comment relates to the diversity measures and how they have been handled. Hill's numbers, like other count, or count-based, measures are affected by sampling effort; all else equal, the greater the sampling effort the greater the diversity, the greater the Hill number. In palaeolimnological studies, sampling effort concerns at least two elements of the data collection and analysis process: i) the sample count, here $\sim$300 valves; and ii) the sedimentation processes, accumulation rates, and depth-based sampling that generated the sediment slices that were counted. The authors appreciate these issues and employed methods to handle varying-effort source 1 via a bootstrap approach.

On L234 the authors state the sediment accumulation rates range between 34 and 133 years per cm, which means that each of the 0.5cm slices contains between 17 and $\sim$66 years. As such, we might expect variation in Hill's numbers simply due to systematic changes in accumulation rates over time. The analyses described here do not appear to account for this element of potential bias due to varying sample effort.

It is unclear to what extent variation in accumulation rates might contribute to or obscure the underlying trends in diversity metrics in the core as accumulation rates are not presented alongside the time series of diversity metrics.

It is also unclear exactly how best to accommodate the varying accumulate rate effect in the rarefaction estimate used by the authors. One option might be to simply scale the results of the rarefaction/bootstrap by the number of years that each sediment sample represents. While simple, this solution might not fully account for the problem if the Chao et al (2014) method makes assumptions about equal effort beyond sample count totals.

A more complicated alternative might be to not use any rarefaction at all, and instead model the observed data using a regression approach, with an offset that includes both the count total and the number of years per slice. For the regression model itself, a GAM seems appropriate given the non-linear change in the metrics over time.

For N0, richness, a Poisson or negative binomial model might be a starting point, and then an offset that is 'log(sample_count * n_years)', where 'sample_count' is the number of valves counted in each sample and 'n_years' is the number of years that each sample represents, should result in a model with the correct normalization to expected number of species per valve count per year. The offset used could be varied so that it is terms of expected richness per 100 valves per decade, by suitable scaling of the values inside the 'log()'.

A similar approach for N1 and N2 is little less clear; Tweedie, gamma, or inverse Gaussian distributions for example can all handle non-negative (Tweedie) or strictly positive (gamma, inverse Gaussian) real values which N1 and N2 result in, and each of these models can include an offset in the same way and, assuming a log link function is used, on the same log scale. But I'm not familiar enough with the details of the gamma and inverse Gaussian models as GLMs to comment further on the exact interpretation of offsets in those models.

The second area I wish to comment on is in the interpretation of the moving window coefficient of variation (CV) results, especially Figure 4. What is being shown in Figure 4? CV is a unit-less variable, but each plot has a label on the y-axis. In the upper panel we see variation in the CV but most of these changes are relatively small suggesting CVs of between 9 and <3 % of the mean. Are these values biologically meaningful? Is the fact that we see CV of log(BVAR) declining in the Late Glacial and early Holocene, really an important decline? The N2 signal seems to show more variation; apart from the Late Glacial period, CV values are between 10% and 50% of the mean N2. Those numbers seem, however, at first sight to indicate that the stability of N2 is much less than log(BVAR). My comment really relates to one of whether palaeo productivity stabilised, and if so, how does that mesh with the interpretation that N2 was also stable for much of the Holocene where the variability in CV values for N2 is so much higher than for BVAR?

The Kruskal Wallis analysis is non-parametric, but it is not without assumptions. The

principal assumption that is certainly violated here is an assumption of independent observations, because your data are a time series. I don't know if there is a way to correct the p-values for the loss of degrees of freedom due to the dependence between samples (as is often done for regression, assuming AR noise), but if not, one might have to use a permutation or resampling/bootstrap procedure where the permutation or resampling is done in a manner that preserves for correlation structure (ruling out simple permutation and simple bootstrap resampling).

My final comment relates to the correlation analysis as shown in Figure 5. What type of correlation are you showing here? All the standard correlation coefficients are bounded -1, 1 but the plot shows values >1. I suspect what you're showing is the coefficients from a linear regression through the scatter of productivity and diversity data in the 1000 year moving windows? This needs to be fully explained in the methods. Putting an informative label on the y-axis would also help the reader understand this figure.

Another problem with Figure 5 is the use of the p values encoded as colours on the plot. It's impossible to tell what the p-values are at the low end, where the interest lies, because of the colour scheme used. Also, you (or the reader) are at risk of making a massive multiple comparisons mistake here. If you are going to use the p values then you would need to correct them for multiple comparisons (i.e. for as many tests as the number of data points shown) using the false discovery rate to adjust the p values. Then you could use a binary indicator perhaps to show which values remain significant.

Other questions remain with this correlation analysis;

* is a linear model in a moving window a good fit to the data?

* if relying on the p values, how biased are these because neither the response nor the covariate are independent,

* if this is a linear regression, which variable played the role of a dependent and independent variables?

* if this was a linear regression why assume all the error is in the dependent variable?

I think it would be better if what was plotted was a true correlation value with bounds at -1 and +1 as that is a scale that most readers will be most familiar with. And in that case you probably wouldn't need to use or present the p-values.

Regarding the principal curves, how confident are you that the data are well represented by a single gradient? The eigenvalues of the CA that you used for the starting curve would be a guide as to whether there is remaining structure in the data on axes 2 and higher. PrCs are really useful when there is a strong single gradient in the data, but when there are secondary gradients my experience is that they are much less useful and can get stuck in some weird solutions that don't make ecological sense. You can get an idea of this by looking at the complexity of the smoothing splines fitted to individual taxa; if these are not simple linear, monotonic, or unimodal curves than it is more likely that the PrC is being asked to do too much and is a good sign of problems with the fit. I mention this because of the very rapid changes in the PrC scores, which can happen when the PrC itself is too complex; it would be useful for the reader to see the 2d ordination with the PrC superimposed on it, perhaps in the supplementary materials?

Finally, throughout there are few attempts to quantify the uncertainties in the quantities you estimate, interpret, discuss, and present to the reader, or to compare the observed results with appropriate null models of no change/trend. This makes it difficult to gauge the overall support of your interpretation that comes from the data and the analyses.

**Minor Comments**

- L189: there's an extra parenthesis before "De'Ath"

- L215: here and throughout, the superscripts in your units appear to have gone missing, perhaps during the conversion to PDF?

- L224: here and throughout, the way you present the Hill's numbers changes throughout the manuscript. Sometimes the "N" is in italics, sometimes not, and sometimes the number is in italics and sometimes not.

- L243: the $R^2$ should be upper case R and the 2 is not superscript

- L294: delete "as" between "declined" and "rapidly" or change the sentence to indicate what BVAR changed as rapidly as.

- L501 add "zone" after "photic"

**References**

Chao, A., Gotelli, N.J., Hsieh, T.C., Sander, E.L., Ma, K.H., Colwell, R.K., Ellison, A.M., 2014. Rarefaction and extrapolation with Hill numbers: a framework for sampling and estimation in species diversity studies. Ecol. Monogr. 84, 45–67. https://doi.org/10.1890/13-0133.1

---

## Referee Comment (RC1) · Anonymous Referee #1 · 7 Jul 2020

General Comments: This is an interesting investigation of the relationships among diatom diversity, diatom productivity (biovolume), and climate stability during the deglaciation and Holocene in Lake Baikal, a lake with limited human influence over this period. The question of how biotic communities responded to climate change or climate variability – in other words their resilience – is of broad interest, particularly given the high-latitude location of the lake and the substantive climate variability on these time scales. The data are of high quality, and the interpretations of trends and relationships are generally well supported. As elaborated below, I suggest only a relatively small number of changes to clarify specific issues or to amplify interpretations.

Specific Comments:

Study site: In the description of the lake, it would be helpful to say something about the primary producers. Specifically, what proportion of the primary producers are diatoms, and how is diatom productivity related to overall primary productivity? What are the primary controls on diatom or primary productivity in the modern lake?

Lines 376-78: It's not clear why you focus only on half of the Holocene Bond Events – why not the intervening ones? If there is a diatom response only to the few Bond Events that you chose here, but not the others, this is worthy of a comment or speculation about why this might be so. Also, these dates are different from those originally proposed by Bond (e.g. he has 5.9 and 4.2 ka, whereas you list 5.2ka) – are these recalibrated? Please clarify.

Line 395-7: So what does it tell us about the environment or about the diatom community if the community has high species richness but low N2 diatom diversity?

Line 400: I'm confused. I thought lines 383-384 say the pre BA community has moderately high species richness, but here you say the flora persists with low richness. Please clarify.

Lines 401-402: Would it make more sense to only consider planktic diatom species richness and diversity? Or also include diversity changes simply among the planktic group? Changes in diversity that are a result of mixing of littoral with planktic communities (taphonomic processes) that didn't really live together is very different from changes of diversity within a single community of species that are actively interacting and competing for resources.

Line 420-422: This raises the general issue about how changes in seasonality might affect diversity, especially given that the diversity of a sample is integrating over multiple decades. If you simply lengthened or shortened the summer season, what would

happen to diversity? I'd like to see a bit more development in the manuscript about the impact of seasonality on diversity, etc.

Line 467-482: I don't think this description of what occurred globally is needed – stick to the global driver (AMOC) and the regional manifestation of climate that drove aquatic change.

Conclusions: I wonder about adding a few thoughts or speculations about how these observations of patterns in Lake Baikal might compare with paleolimnological observations/reconstructions made in other regions or from other kinds of systems about diatom resilience to natural climate variability (for example, Jovanovska et al. 2016; Benito et al. 2019). There have been a few recent papers out on this. This would emphasize the broader significance of the results.

Technical comments:

Line 322, 331: Northern Hemisphere is a formal and specific geographic place – I was taught it should be capitalized (but maybe that is a US English thing?).

Line 447: replace "are" with "is"

Lines 540, 541: in both lines a ")" is missing

---

## Referee Comment (RC2) · Anonymous Referee #2 · 16 Oct 2020

Review of "Long term trends in aquatic diversity, productivity and stability: a 15,800 year multidecadal diatom study from Lake Baikal, southern Siberia" by Mackay et al. for Climate of the Past – Discussions. This manuscript presents a high resolution diatom record (ca. six-decade) of Termination 1 and the Holocene and uses diversity, species abundances, and paleoproductivity measures to explore the relationships among diversity, resilience, and stability change as a result of climate drives over the last 15800 yrs. The manuscript is well-written and conceived, reaches solid conclusions, and I recommend its acceptance following minor corrections and revisions as

discussed below. Nice paper.

Broader comments:

Figures – several of the figures are wanting for reproducibility or interpretibility. If they are to be produced at the size provided in the review copy, they are unacceptable for publication. A reader should not have to get a hand lens out to interpret a figure. This is especially apparent in Figs 2, 3, 6. A few other notes, what is meant by the (agg) in Aulacoseira skvortzowii in Fig 3, meyeri misspelled in Fig 3, units on seasonal BVAR does not use a mu symbol for $\mu$m and the $\mu$m3 and cm2 do not have superscripted exponents in Fig. 3. In Figure 4 and 5 the record is truncated at aout 14500 yBP. The core is clearly shown to be 15800 yBP old, why the truncated records in Figs 4, 5?

M&M – the section on diatom analysis is strangely variable in detail. Diatoms are described for a general audience, but then there analysis is described as though everyone knows how they are treated in sediment analysis. For example, what is meant by 5 mm resolution? Valves per gram of what. Add that they "possess a silica shell called valves. . ."

Taxonomic names – care should be taken to make sure taxonomic names are correctly spelled and formatted throughout the manuscript. Stephanodiscus meyeri (single -i at end), the v. in variety radians should be in Roman font, not italicized.

Discussion – has there been similar detailed approaches taken on other long records? This seems to be a novel approach for considering the relations between diversity/stability and climate, but that aspect is not highlighted by the authors and it should be! Has this approach of melding resource ecology and diversity been applied to other climate records, perhaps from varved lakes and accounting for Holocene scale records (LIA, MCA, HCM)?

Discussion – earlier efforts by Khursevich et al. (2001, 2005) and Edlund (2006) have considered the longer Baikal records, but in lower resolution and and with fewer measures of diversity and productivity. How does this new record compare or contrast with those earlier approaches to examine the full Baikal record. Is the Pleistocene/Holocene transition unique? For example, this paper suggests higher valve flux and BVAR in glacial times (T1) vs Holocene (see Table 1). This seems to contrast with many of the diatom depauperate regions characteristic of other glacial periods in Baikal's history.

Discussion – the Baikal diatom community is characterized by high endemicity. Is there any reason to believe that this endemic flora drives the patterns shown in your data, i.e. is the resiliency a byproduct of endemicity? ln 521-533

Minor corrections ln 52, understanding…is ln 117, remove comma after "…2018) restricted…" ln 130, change to "…diveristy that is not experienced…" ln 133, change to "…events disrupt these …" ln 140/41, odd expression outside of UK, change to "…due to its diverse flora..." ln 142, provide refernce to endemicity of Baikal ln 145, italicize the ship's name. She deserves that. ln 149-50, add space before meter abbreviation in 3 places. Check rest of msc for same. ln 160/61, superscript 14 in 14C, check rest of msc for same, also noted for $\mu$m3 (ln 215), etc. ln 222, clarify what is meant by PDR? Is PDR the relationship between paleoproductivity and N2 or is it N2? If it is the relationship, how is it calculated? Fig 5 seems to be reporting this "PDR" but PDR is not described or connected to Fig 5. What am I missing as a reader here? Is Fig 6b also related to PDR? ln 399, "…Bolling, the pre-Bolling diatom…" ln 415, why is the there a delay in the N2 diversity decline? Would be worth some speculation. ln 456, skvortzowii misspelled ln 475, close parenthesis after 2011) ln 475, verb agreement "shows" ln 509, check msc for formatting of N2, N0, N1, italicized N and Roman 2 seems the standard. ln 541, close parenthesis after Fig. 3)

---

## Author Comment (AC1) · 3 Feb 2021

Dr Gavin Simpson comments

1. My first comment relates to the diversity measures and how they have been handled. Hill's numbers, like other count, or count-based, measures are affected by sampling effort; all else equal, the greater the sampling effort the greater the diversity, the greater the Hill number. In palaeolimnological studies, sampling effort concerns at least two elements of the data collection and analysis process: i) the sample count,

here _300 valves; and ii) the sedimentation processes, accumulation rates, and depth-based sampling that generated the sediment slices that were counted. The authors appreciate these issues and employed methods to handle varying-effort source 1 via a bootstrap approach... It is unclear to what extent variation in accumulation rates might contribute to or obscure the underlying trends in diversity metrics in the core as accumulation rates are not presented alongside the time series of diversity metrics... A more complicated alternative might be to not use any rarefaction at all, and instead model the observed data using a regression approach, with an offset that includes both the count total and the number of years per slice. For the regression model itself, a GAM seems appropriate given the non-linear change in the metrics over time.

Authors Response • Dr Simpson is essentially asking how can we account for variable SARs (which vary by a factor of 4) in interpreting Hill's numbers for species richness / diversity, and he helpfully provides a number of suggestions to consider. We agree that we should take into account SARs when presenting and discussing our diversity measures. Therefore, in the main body of the manuscript we will present new analyses for rarefied Hills diversity measures N0, N1 and N2 divided by the accumulation rate over time. • Note that we have also taken the opportunity to update our age modelling using IntCal20, although specific dates over our timeframe are similar to those obtained from using IntCal13.

• We have also looked at Dr Simpson's suggestion of the more complicated alternative, calculating Hill's diversity measures without randomization and using the predicted values of a GAM model specified as gam.rich1 <- gam(N0 ∼ s(age, k = 70), offset = log(counts*AccumRate), family = poisson, method = "REML", data = diversity). These will be presented in Supplementary Info, alongside a brief discussion of its merits and differences with data presented in the main paper.

2. The second area I wish to comment on is in the interpretation of the moving window coefficient of variation (CV) results, especially Figure 4. What is being shown in Figure 4? CV is a unit-less variable, but each plot has a label on the y-axis. In the upper panel

we see variation in the CV but most of these changes are relatively small suggesting CVs of between 9 and <3 % of the mean. Are these values biologically meaningful? Is the fact that we see CV of log(BVAR) declining in the Late Glacial and early Holocene, really an important decline? The N2 signal seems to show more variation; apart from the Late Glacial period, CV values are between 10% and 50% of the mean N2. Those numbers seem, however, at first sight to indicate that the stability of N2 is much less than log(BVAR). My comment really relates to one of whether palaeo productivity stabilised, and if so, how does that mesh with the interpretation that N2 was also stable for much of the Holocene where the variability in CV values for N2 is so much higher than for BVAR?

Authors Response • Dr Simpson's concerns here relate to (i) whether the interpretations we make with respect to changing CV over time are ecologically meaningful for log(BVAR) (as variation is rather minor) and has palaeoproductivity really stabilised during the Holocene; (ii) we originally interpreted N2 as being rather stable for the Holocene, yet higher variability for N2 is counter to that interpretation.

• We note that Dr Simpson doesn't object to the use of CV, but had queries about interpretations surrounding stability. We accept that CV will be a rather crude measure of stability, and we will reanalyse CV for N2 data taking into account variable SARs, as undertaken above.

3. The Kruskal Wallis analysis is non-parametric, but it is not without assumptions. The principal assumption that is certainly violated here is an assumption of independent observations, because your data are a time series. I don't know if there is a way to correct the p-values for the loss of degrees of freedom due to the dependence between samples (as is often done for regression, assuming AR noise), but if not, one might have to use a permutation or resampling/bootstrap procedure where the permutation or resampling is done in a manner that preserves for correlation structure (ruling out simple permutation and simple bootstrap resampling).

Authors Response âǍć We agree with Dr Simpson's comments and will delete the test of whether total, spring and autumn productivity, and N2 diversity were different for the three centuries immediately before and after the three Holocene climate disturbance events using the non-parametric Kruskal-Wallis one-way analysis of variance (Table 2).

4. My final comment relates to the correlation analysis as shown in Figure 5. What type of correlation are you showing here? All the standard correlation coefficients are bounded -1, 1 but the plot shows values >1. I suspect what you're showing is the co-efficients from a linear regression through the scatter of productivity and diversity data in the 1000 year moving windows? This needs to be fully explained in the methods. Putting an informative label on the y-axis would also help the reader understand this figure. Another problem with Figure 5 is the use of the p values encoded as colours on the plot. It's impossible to tell what the p-values are at the low end, where the interest lies, because of the colour scheme used. Also, you (or the reader) are at risk of making a massive multiple comparisons mistake here. If you are going to use the p values then you would need to correct them for multiple comparisons (i.e. for as many tests as the number of data points shown) using the false discovery rate to adjust the p values.

5. Then you could use a binary indicator perhaps to show which values remain sig-nificant. Other questions remain with this correlation analysis: is a linear model in a moving window a good fit to the data?; if relying on the p values, how biased are these because neither the response nor the covariate are independent; if this is a linear re-gression, which variable played the role of a dependent and independent variables?

Authors Response âǍć These comments were perhaps the most substantial for us to deal with as they go to the heart of what we are trying to get out from our data. As pointed out, one of the problems of the moving window analysis was that it was highly subjective to the size of the window, and Dr Simpson was right in that sometimes a linear model might not be the best model to describe the data.

âǍć We have decided instead to take a different approach to investigate productivitydiversity relationships over time, which is hypothesis driven and, we believe, more robust. Through independent palaeoclimate studies, we have identified three periods of climate change in southern Siberia over the past 15,000 years. A period of rapid warming (the Bølling-Allerød), cooling (the Younger Dryas) and a period of relative stability (mid-Holocene). Using a non-linear approach, we will test productivity-diversity relationships.

6. Regarding the principal curves, how confident are you that the data are well represented by a single gradient? The eigenvalues of the CA that you used for the starting curve would be a guide as to whether there is remaining structure in the data on axes 2 and higher. PrCs are really useful when there is a strong single gradient in the data, but when there are secondary gradients my experience is that they are much less useful and can get stuck in some weird solutions that don't make ecological sense. You can get an idea of this by looking at the complexity of the smoothing splines fitted to individual taxa; if these are not simple linear, monotonic, or unimodal curves than it is more likely that the PrC is being asked to do too much and is a good sign of problems with the fit. I mention this because of the very rapid changes in the PrC scores, which can happen when the PrC itself is too complex; it would be useful for the reader to see the 2d ordination with the PrC superimposed on it, perhaps in the supplementary materials?

Authors Response • As suggested by Dr Simpson, we checked the result of the CA. The variation capture by axis 1 was only explaining ∼4% of the total inertia/variation. The first four axes seem relevant (in total explaining a little bit more than 11% of the total variation), so based on this actually PrC might not be a very good choice in this case.

• Comments from Reviewer 1, Point #5 have already persuaded us that we need to reanalyse our data based on separating out planktonic taxa from benthic taxa, as multiple gradients indicated by the PrCs may be, for example, reflective of the different habitats.

7. Finally, throughout there are few attempts to quantify the uncertainties in the quantities you estimate, interpret, discuss, and present to the reader, or to compare the observed results with appropriate null models of no change/trend. This makes it difficult to gauge the overall support of your interpretation that comes from the data and the analyses

Authors Response • We think through a more focussed, hypotheses-driven approach our interpretations that come from the data and the analyses will be more robust. We will ensure that uncertainties are considered throughout.

8. Minor Comments- L189: there's an extra parenthesis before "De'Ath"; - L215: here and throughout, the superscripts in your units appear to have gone missing, perhaps during the conversion to PDF?; - L224: here and throughout, the way you present the Hill's numbers changes through out the manuscript. Sometimes the "N" is in italics, sometimes not, and sometimes the number is in italics and sometimes not; - L243: the RÊĘ2 should be upper case R and the 2 is not superscript; - L294: delete "as" between "declined" and "rapidly" or change the sentence to indicate what BVAR changed as rapidly as; - L501 add "zone" after "photic"

Authors Response • These have all been taken account of. For information, all Hill's numbers are given with N italicised, and the following number not, i.e. N0, N1 and N2 (as per Felde et al. 2016).

---

## Author Comment (AC2) · 3 Feb 2021

1. In the description of the lake, it would be helpful to say something about the primary producers. Specifically, what proportion of the primary producers are diatoms, and how is diatom productivity related to overall primary productivity? What are the primary controls on diatom or primary productivity in the modern lake?

Authors Response:  c We will add the following sentences to the section "Study site", with appropriate references.

[Figure]

"Diatoms comprise between 50-90% of phytoplankton biomass during the Spring bloom under ice and after ice break-up (Popovskaya et al. 2015; Panizzo et al. 2017), that contributes a significant proportion of overall annual primary productivity (Popovskaya 2000). With the seasonal onset of summer warming and surface water stratification, diatoms decline in importance, and are replaced by non-siliceous autotrophic picoplankton (Fietz et al. 2005; Belykh et al. 2006). During autumn turnover, a smaller diatom bloom once more dominates primary production. Nitrogen and phosphorus co-limit photic zone productivity in Lake Baikal (Satoh et al. 2006; O'Donnell et al. 2017), with rates of deep-water nutrient supply increasing markedly since the mid 19th century (Swann et al. 2020)."

2. Lines 376-78: It's not clear why you focus only on half of the Holocene Bond Events – why not the intervening ones? If there is a diatom response only to the few Bond Events that you chose here, but not the others, this is worthy of a comment or speculation about why this might be so. Also, these dates are different from those originally proposed by Bond (e.g. he has 5.9 and 4.2 ka, whereas you list 5.2ka) – are these recalibrated? Please clarify.

Authors Response • The reviewer makes a good point here, in that by focussing on only selected Bond Events we are in effect biasing our interpretation towards only events where we have a coincident signal. As we have been afforded the opportunity to re-look and reanalyse our data, based on other reviewers comments, we will instead focus on known periods of global climate change (both abrupt and slow), so that we can investigate relationships between the different biological response variables under different climate scenarios. We will therefore focus our efforts on the more variable (i.e. the Bølling-Allerød interstadial and the Younger Dryas stadial) and more stable (the mid- Holocene) climatic periods. This also takes away from any dating uncertainty about when Bond events occurred in the record.

3. Line 395-7: So what does it tell us about the environment or about the diatom community if the community has high species richness but low N2 diatom diversity?

Author's Response • The time period indicated here is during the latter stages of Termination 1, where a peak in diatom richness (N0) is coincident with a peak in diatom N2 diversity, followed by drops in both. What we can say is that higher diatom richness closely follows the patterns in total benthic diatoms, while in general N2 diversity reflects dominant taxa in the record. As we are now presenting diversity data just for the plankton in Lake Baikal (see comment #5 below), we expect richness and diversity relationships to change somewhat from the original manuscript.

4. Line 400: I'm confused. I thought lines 383-384 say the pre BA community has moderately high species richness, but here you say the flora persists with low richness. Please clarify

Author's Response • Line 383-384 refers to a comparison of diatom richness between Termination 1 and the Holocene, for which there was no significant difference between the two time periods, with N0 = 22.7 and 24.1 respectively. In Line 400, we agree that our use of "pre- Bølling diatom flora persists, with concomitant low richness" is only correct for part of the Termination 1 record, as clearly there is a short-lived peak in richness as well. We have edited Lines 399-400 and have deleted ". . .with concomitant low richness"

5. Lines 401-402: Would it make more sense to only consider planktic diatom species richness and diversity? Or also include diversity changes simply among the planktic group? Changes in diversity that are a result of mixing of littoral with planktic communities (taphonomic processes) that didn't really live together is very different from changes of diversity within a single community of species that are actively interacting and competing for resources.

Author's Response • We agree with the reviewer's comments here, and will calculated diversity and richness based on diatoms identified to be planktonic from the key Lake Baikal flora (Popovskaya & Likhoshvai 2011), to simplify these interpretations, and to reflect that the fact that benthic diatoms occupy completely different habitats are

so are not competing for the same resources.

6. Line 420-422: This raises the general issue about how changes in seasonality might affect diversity, especially given that the diversity of a sample is integrating over multiple decades. If you simply lengthened or shortened the summer season, what would happen to diversity? I'd like to see a bit more development in the manuscript about the impact of seasonality on diversity, etc.

Author's Response  c Our original interpretations were based on diversity records of combined planktonic and benthic diatoms. With reanalyses of diversity measures based only on planktonic and tychoplanktonic taxa, our interpretations will change somewhat throughout the discussion. However, we take on board the reviewer's comment re. the influence of seasonality, and will certainly consider further.

7. Line 467-482: I don't think this description of what occurred globally is needed – stick to the global driver (AMOC) and the regional manifestation of climate that drove aquatic change.

Author's Response  c We agree, and we will make this and other sections (which were overly focussed on selected periods of abrupt change coincident with Bond events) much more succinct, or we will remove them altogether.

8. Conclusions: I wonder about adding a few thoughts or speculations about how these observations of patterns in Lake Baikal might compare with paleolimnological observations/ reconstructions made in other regions or from other kinds of systems about diatom resilience to natural climate variability (for example, Jovanovska et al. 2016; Benito et al. 2019). There have been a few recent papers out on this. This would emphasize the broader significance of the results.

Authors Response  c These are interesting suggestions, and will consider how best to incorporate them in the conclusions.

9. Technical comments

Authors Response • All three amendments have been made

References used in this report, and now added to the manuscript

Belykh, O. I., Ekaterina, G., Sorokovikova, T., SaphonovaI. A., Tikhonova, V: Autotrophic picoplankton of Lake Baikal: Composition, abundance and structure. Hydrobiol., 568, 9-17, 2006

O'Donnell, D. R., Wilburn, P., Silow, E. A., Yampolsky, L. Y., Litchman, E: Nitrogen and phosphorus colimitation of phytoplankton in Lake Baikal: insights from a spatial survey and nutrient enrichment experiments. Limnol. Oceanogr. 62, 1383-1392, 2017.

Panizzo, V. N., Swann, G. E. A., Mackay, A. W., Vologina, E., Alleman, L., Andre, L., Pashley, V. H., Horstwood, M. S. A: Constraining modern day silicon cycling in Lake Baikal. Global. Biogeochem. Cy. 31, 556-574, 2017.

Popovskaya, G. I.: Ecological monitoring of phytoplankton in Lake Baikal. Aquat. Ecosyst. Health. 3, 215-225. 2000.

Popovskaya, G. I., Likhoshvai, E. V.: Plankton Diatom Algae in Lake Baikal: Key Atlas, Novosibirsk: Nauka, 2011. [in Russian].

Popovskaya, G. I. Usol'tseva, M. V. Domysheva, V. M. Sakirko, M. V. Blinov, V. V. Khodzher, T. V: The spring phytoplankton in the pelagic zone of Lake Baikal during 2007- 2011. Geogr. Nat. Resour. 36, 253-262, 2015.

Satoh, Y. Katano, T. Satoh, T, Mitamura, O. Anbutsu, K. Nakano, S. Ueno, H. Kihira, M. Drucker, V. Tanaka, Y. Mimura, T. Watanabe, Y. Sugiyama, M: Nutrient limitation of the primary production of phytoplankton in Lake Baikal. Limnology 7, 225-229, 2006.

Swann, G. E. A, Panizzo, V. N., Piccolroaz, S., Pashley, V., Horstwood, M. S. A., Roberts, S., Vologina, E., Piotrowska, N., Sturm, M., Zhdanoc, A., Granin, N., Normal, C., McGowan, S., Mackay, A. W: Changing nutrient cycling in Lake Baikal: the world's oldest lake. Proc. Nat. Acad. Sci., 117, 27211-27217, 2020.

---

## Author Comment (AC3) · 3 Feb 2021

**Reviewer 2 comments**

10. Figures – several of the figures are wanting for reproducibility or interpretability. If they are to be produced at the size provided in the review copy, they are unacceptable for publication. A reader should not have to get a hand lens out to interpret a figure. This is especially apparent in Figs 2, 3, 6. A few other notes, what is meant by the (agg) in Aulacoseira skvortzowii in Fig 3, meyeri misspelled in Fig 3, units on seasonal BVAR

does not use a mu symbol for \_m and the \_m3 and cm2 do not have superscripted exponents in Fig. 3. In Figure 4 and 5 the record is truncated at aout 14500 yBP. The core is clearly shown to be 15800 yBP old, why the truncated records in Figs 4, 5?

Authors Response ć The review makes important points re. reproducibility & interpretability of some of the figures. These figures will have to been redrawn anyway to make way for changes in recommended new analyses, and we we ensure that all labels etc meet requirements for CP. Other minor corrections have been made, and Figs 4 and 5 no longer exist.

11. M&M – the section on diatom analysis is strangely variable in detail. Diatoms are described for a general audience, but then there analysis is described as though everyone knows how they are treated in sediment analysis. For example, what is meant by 5 mm resolution? Valves per gram of what. Add that they "possess a silica shell called valves: : :"

Authors Response ć We have adapted the methods section for diatoms, to take account of these concerns: Lines 197- onwards.

12. Taxonomic names – care should be taken to make sure taxonomic names are correctly spelled and formatted throughout the manuscript. Stephanodiscus meyeri (single -i at end), the v. in variety radians should be in Roman font, not italicized

Authors Response ć We have corrected the spelling for S. meyeri, and unitalicized v.

13. Discussion – has there been similar detailed approaches taken on other long records? This seems to be a novel approach for considering the relations between diversity/ stability and climate, but that aspect is not highlighted by the authors and it should be! Has this approach of melding resource ecology and diversity been applied to other climate records, perhaps from varved lakes and accounting for Holocene scale records (LIA, MCA, HCM)?

CPD
Authors Response ć As far as we are aware, there are few detailed records as ours, hence the novelty and value of this study. And there has not, as far as we are aware, been other studies which have looked at relations between diversity/ stability and climate over such long timescales. We think we highlight this novelty in the Introduction, Lines 96-100 "This constitutes an important gap in our knowledge because in terms of climate change, PDR and resource use efficiency (Gross and Cardinale 2007; Ptacnik et al. 2008) will be fundamentally different over long (e.g. climate and landscape evolution) and short (e.g. pulse disturbances such as climate disturbance events (Kéfi et al. 2019)) timescales."

ć But we will emphasise the novelty of this record again in the Discussion

14. Discussion – earlier efforts by Khursevich et al. (2001, 2005) and Edlund (2006) have considered the longer Baikal records, but in lower resolution and with fewer measures of diversity and productivity. How does this new record compare or contrast with those earlier approaches to examine the full Baikal record. Is the Pleistocene/Holocene transition unique? For example, this paper suggests higher valve flux and BVAR in glacial times (T1) vs Holocene (see Table 1). This seems to contrast with many of the diatom depauperate regions characteristic of other glacial periods in Baikal's history

Authors Response ć The reviewer makes a really useful suggestion in comparing our records to previous studies done on Lake Baikal, albeit at lower resolution. We will add a new section (see below) which brings in the three studies that they highlight, plus Bradbury et al. 1994, which covers a very similar timescale as our own, with cores from off the shoulder of the Selenga Delta spanning the past 15,000 years or so. Thank you.

New section to be added: 4.4 Comparison of Vydrino record with other long diatom sequences from Lake Baikal

It has long been recognised that Quaternary biogenic silica and diatom concentrations in Lake Baikal sediments mirror changes in insolation (Khursevich et al. 2001), such that very low concentrations characterise glacial periods, likely due to a number of Interactive comment

factors including lower productivity and higher rates of diatom dissolution as well as dilution due to increased inputs of clastic material (see Mackay 2007 for a review). The Vydrino late glacial – Holocene record has an almost identical diatom assemblage to those identified for the same time period in long cores extracted from both the Posol-skaya Bank (BDP-99) and Academician Ridge (BDP-96-2) to the north (Khursevich et al. 2005) (Fig. 1). In another study of Quaternary Lake Baikal diatoms, this time from the Buguldieka Saddle (Fig. 1), Edlund (2006) found that although earlier glaciations also contained few remaining diatoms, the "Sartan glaciation", i.e. MIS2, still contained at least 10 species of planktonic taxa, and an assemblage again very similar to our Vydrino sequence. Bradbury et al. (1994) produced a similar but much lower resolution record for diatom changes spanning the past 15,000 years from the Posolskyaya Bank, where both the assemblage and sequence of diatoms are similar to Vydrino. Therefore, observations and conclusions drawn in this study related to productivity diversity relationships are likely applicable to other regions of this vast lake at least during the same time period.

15. Discussion – the Baikal diatom community is characterized by high endemicity. Is there any reason to believe that this endemic flora drives the patterns shown in your data, i.e. is the resiliency a by-product of endemicity? In 521-533

Authors Response ć This is a very interesting question. Rather than the endemicity per se, it's likely the length of time spanned by the continuous sedimentary records found in ancient lakes that is the primary factor promoting resiliency. Of course, as endemicity needs a lot of time to develop, ancient lakes have generally high endemicity. We will include this concept in our discussion, drawing in work from the recently published paper by Luethje & Snyder (2021, Phytotaxa) in which they discuss how climate events appear to drive morphological transition within a "species complex" (in that case Pantocsekiella, i.e. the species around the "Cyclotella ocellata/comensis complex) on a long time-scale (0 to i.2 Ma) in Lake El'gygytgyn. In their study, they identify a morphodeme as a distinct species, partly on the basis that its occurrence in the record
corresponds with stable climatic conditions during a 180 ka-long interglacial (when the lack of competition from other diatoms allowed the speciation of this large-sized morphodeme). The idea is, given sufficient time and opportunity (i.e. the demise of a competitor for a particular niche), morphological variants of a species (in their case, a large-celled variant) will be allow to thrive and occupy the vacant niche. The Cyclotella (Lindavia) baicalensis/ornata/minuta complex in Lake Baikal is another example of speciation as the different sizes allow them to occupy different habitat/seasonal niches.

16. Minor corrections: In 52, understanding: : :is In 117, remove comma after ": : :2018) restricted: : :" In 130, change to ": : :diveristy that is not experienced: : :" In 133, change to ": : :events disrupt these : : :" In 140/41, odd expression outside of UK, change to ": : :due to its diverse flora..." In 142, provide reference to endemicity of Baikal In 145, italicize the ship's name. She deserves that. In 149-50, add space before meter abbreviation in 3 places. Check rest of msc for same. In 160/61, superscript 14 in 14C, check rest of msc for same, also noted for \_m3 (In 215), etc. In 222, clarify what is meant by PDR? Is PDR the relationship between paleoproductivity and N2 or is it N2? If it is the relationship, how is it calculated? Fig 5 seems to be reporting this "PDR" but PDR is not described or connected to Fig 5. What am I missing as a reader here? Is Fig 6b also related to PDR? In 399, ": : :Bolling, the pre-Bolling diatom: : :"

Authors Response ć We have taken all of these onboard

17. In 415, why is there a delay in the N2 diversity decline? Would be worth some speculation.

Authors Response ć We actually speculate on this from lines 418... "The lag in N2 diversity decline suggests that available resources for diatom growth were not initially limiting; species composition at this time is dominated by the spring blooming S. acus v. radians and the autumnal blooming C. inconspicua (Fig. 3h,k), and therefore these species may not be directly competing for the same resources (Interlandi & Kilham 2001)."
18. Ln 456, skvortzowii misspelled ln 475, close parenthesis after 2011) ln 475, verb agreement "shows" ln 509, check msc for formatting of N2, N0, N1, italicized N and Roman 2 seems the standard. In 541, close parenthesis after Fig. 3)

Authors Response ć We have corrected these errors.

References used in this report, and now added to the manuscript

Bradbury, J. P., Bezrukova, Ye. V., Chernyaeva, G. P., Colman, S. M., Khursevich, G., King, J. W., Likoshway, Ye. V.: A synthesis of post-glacial diatom records from Lake Baikal. J. Paleolim., 10, 213–252, 1994.

Edlund, M. B.: Persistent low diatom plankton diversity within the otherwise highly diverse Lake Baikal ecosystem. Nova Hedwigia 1310, 339–356, 2006.

Khursevich, G. K., Karabanov, E. B., Prokopenko, A. A., Williams, D. F., Kuzmin, M. I., Fedenya, S. A., Gvozdkov, A. A.: Insolation regime in Siberia as a major factor controlling diatom production in Lake Baikal during the past 800,000 years. Quat. Int. 80–81, 47–58, 2001.

Khursevich, G. K., Prokopenko, A. A., Fedenya, S. A., Tkachenko, L. I., Williams, D. F.: Diatom biostratigraphy of Lake Baikal during the past 1.25 Ma: new results from BDP-96-2 and BDP-99 drill cores. Quat. Int., 136, 95–104, 2005.

Luethje, M., Snyder, J.,: Climate-related morphological changes in Pantocsekiella (Mediophyceae) spanning 0-1.2 Ma in the Lake El'gygytgyn, northeastern Russia including Pantocsekiella elgygytgynensis sp. nov. Phytotaxa, 478, 67-91, 2021.

**CPD**

---

## Author Response (AR1)

Dear Dr Mills,

I'd like to take the opportunity to thank you and the editorial team for providing us with so much flexibility to undertake these corrections, in light of my own health matters and of my of my co-authors. This genuinely minimised stress for both myself and Dr Felde, and has ultimately led to a much improved paper.

Below we detail our responses and comments to all three reviewers. Their comments were pertinent and have helped improve the manuscript at every level, allowing us to put forward a tighter and more robust set of analyses to test better-defined hypotheses. Major changes in summary are:

- The diatom data themselves were recalculated and reanalysed, to include only species that are adapted to grow in the open pelagic lake
- The age model has been updated to take account of the latest calibration (IntCal20).
- Richness and diversity data have been recalculated taking into account sediment accumulation rates
- Instead of taking a rather biased view of only discussing Bond events that show a signal, we have instead used independent climate periods determined from the literature to form the framework for our analyses. We used these climate periods to hypothesise how, for example, productivity – diversity relationships may differ during terms of periods of relative climate instability and periods of relative climate stability
- At reviewers recommendation we have also undertaken new analyses to show differences among the climate periods using generalized additive models (GAM) and classification regression trees (CRT).
- Our other quantitative analyses have been omitted

With best wishes,

Anson (on behalf of all the authors)

**Reviewer 1 comments**

1.  *In the description of the lake, it would be helpful to say something about the primary producers. Specifically, what proportion of the primary producers are diatoms, and how is diatom productivity related to overall primary productivity? What are the primary controls on diatom or primary productivity in the modern lake?*

**Authors Response:**
- We have added the following sentences to the section "Diatom analysis", with appropriate references.

"*Diatoms comprise between 50-90% of phytoplankton biomass during the Spring bloom under ice and after ice break-up (Popovskaya et al. 2015; Panizzo et al. 2017), that contributes a significant proportion of overall annual primary productivity (Popovskaya 2000). With the seasonal onset of summer warming and surface water stratification, diatoms decline in importance, and are replaced by non-siliceous autotrophic picoplankton (Fietz et al. 2005; Belykh et al. 2006). During autumn turnover, a smaller diatom bloom once more dominates primary production. Nitrogen and phosphorus co-limit photic zone productivity in Lake Baikal (Satoh et al. 2006; O'Donnell et al. 2017), with rates of deep-water nutrient supply increasing markedly since the mid 19th century (Swann et al. 2020).*"

2.  *Lines 376-78: It's not clear why you focus only on half of the Holocene Bond Events – why not the intervening ones? If there is a diatom response only to the few Bond Events that you chose here, but not the others, this is worthy of a comment or speculation about why this might be so. Also, these dates are different from those originally proposed by Bond (e.g. he has 5.9 and 4.2 ka, whereas you list 5.2ka) – are these recalibrated? Please clarify.*

**Authors Response**
- The reviewer makes a good point here, in that by focussing on only selected Bond Events we are in effect biasing our interpretation towards only events where we have a coincident signal.
- As we have been afforded the opportunity to re-look and reanalyse our data, based on other reviewers comments, we have instead focus on known periods of global climate change, so that we can investigate relationships between the different biological response variables under different climate scenarios.
- We have therefore refocused our efforts to explore trends in diatom diversity, palaeoproductivity and productivity – diversity relationships among independent and pre-defined climate periods: the pre Bølling-Allerød, the Bølling-Allerød interstadial, the Younger Dryas stadial, and the now three official stages of the Holocene (Early, Middle, Late) (Walker et al. 2018). We have also undertaken new analyses to show differences among the climate periods using generalized additive models (GAM) and multivariate classification trees (MCT).

3.  *Line 395-7: So what does it tell us about the environment or about the diatom community if the community has high species richness but low N2 diatom diversity?*

**Author's Response**
- The time period indicated here is during the latter stages of Termination 1, where a peak in diatom richness ($N0$) is coincident with a peak in diatom $N2$ diversity, followed by drops in both. What we can say is that higher diatom richness closely follows the patterns in total benthic diatoms, while in general $N2$ diversity reflects dominant taxa in the record.
- However, as we are now presenting diversity data just for the planktonic and tychoplanktonic diatoms in Lake Baikal (see comment #5 below), richness and diversity relationships have changed in places from the original manuscript. Moreover, at the recommendation of

Simpson, diversity and richness are plotted taking into account sediment accumulation rates as well.

4. *Line 400: I'm confused. I thought lines 383-384 say the pre BA community has moderately high species richness, but here you say the flora persists with low richness. Please clarify*

**Author's Response**
- Line 383-384 referred to a comparison of diatom richness between Termination 1 and the Holocene, for which there was no significant difference between the two time periods, with $N0 = 22.7$ and 24.1 respectively. In Line 400, we agree that our use of *"pre- Bølling diatom flora persists, with concomitant low richness"* is only correct for part of the Termination 1 record, as clearly there is a short-lived peak in richness as well. We have edited Lines 399-400 and have deleted "…*with concomitant low richness"*
- However, much of the text has been re-written taking into account that we have reanalysed our diversity data.

5. *Lines 401-402: Would it make more sense to only consider planktic diatom species richness and diversity? Or also include diversity changes simply among the planktic group? Changes in diversity that are a result of mixing of littoral with planktic communities (taphonomic processes) that didn't really live together is very different from changes of diversity within a single community of species that are actively interacting and competing for resources.*

**Author's Response**
- We agree with the reviewer's comments here, and we have now recalculated diversity and richness based on diatoms identified to be planktonic and tychoplanktonic from the key Lake Baikal flora (Popovskaya & Likhoshvai 2011). This has allowed us to simplify our interpretations, and to reflect that the fact that benthic diatoms occupy completely different habitats are so are not competing for the same resources.

6. *Line 420-422: This raises the general issue about how changes in seasonality might affect diversity, especially given that the diversity of a sample is integrating over multiple decades. If you simply lengthened or shortened the summer season, what would happen to diversity? I'd like to see a bit more development in the manuscript about the impact of seasonality on diversity, etc.*

**Author's Response**
- Our original interpretations were based on diversity records of combined planktonic and benthic diatoms. With reanalyses of diversity measures based only on planktonic and tychoplanktonic taxa, our interpretations reflect patterns in the revised dataset.
- We also take on board the reviewer's comment re. the influence of seasonality, especially when periods of ice cover are longer on the lake, and have incorporated relevant interpretations throughout the text. We have also plotted on Fig. 8, changes in obliquity which help with interpreting changes according to seasonality in this region of central Asia.

7. *Line 467-482: I don't think this description of what occurred globally is needed – stick to the global driver (AMOC) and the regional manifestation of climate that drove aquatic change.*

**Author's Response**
- We agree, and have made this and other sections (which were overly focussed on selected periods of abrupt change coincident with Bond events) much more succinct.

8. *Conclusions: I wonder about adding a few thoughts or speculations about how these observations of patterns in Lake Baikal might compare with paleolimnological observations/ reconstructions made in other regions or from other kinds of systems about diatom resilience to natural climate*

*variability (for example, Jovanovska et al. 2016; Benito et al. 2019). There have been a few recent papers out on this. This would emphasize the broader significance of the results.*

**Authors Response**
- These are interesting suggestions, and we have used Jovanovska et al. 2016 in our conclusions.

9. *Technical comments*

**Authors Response**
- All three amendments have been made

**Reviewer 2 comments**

10. *Figures – several of the figures are wanting for reproducibility or interpretability. If they are to be produced at the size provided in the review copy, they are unacceptable for publication. A reader should not have to get a hand lens out to interpret a figure. This is especially apparent in Figs 2, 3, 6. A few other notes, what is meant by the (agg) in Aulacoseira skvortzowii in Fig 3, meyeri misspelled in Fig 3, units on seasonal BVAR does not use a mu symbol for _m and the _m3 and cm2 do not have superscripted exponents in Fig. 3. In Figure 4 and 5 the record is truncated at about 14500 yBP. The core is clearly shown to be 15800 yBP old, why the truncated records in Figs 4, 5?*

**Authors Response**
- The review makes important points re. reproducibility & interpretability of the figures. These figures have been redrawn to make way for changes in recommended new analyses, and we checked that all labels etc meet requirements for CP. Other minor corrections have been made, and Figs 4 and 5 from original manuscript no longer exist.

11. *M&M – the section on diatom analysis is strangely variable in detail. Diatoms are described for a general audience, but then there analysis is described as though everyone knows how they are treated in sediment analysis. For example, what is meant by 5 mm resolution? Valves per gram of what. Add that they "possess a silica shell called valves: : :"*

**Authors Response**
- We have adapted the methods section for diatoms, to take account of these concerns

12. *Taxonomic names – care should be taken to make sure taxonomic names are correctly spelled and formatted throughout the manuscript. Stephanodiscus meyeri (single -i at end), the v. in variety radians should be in Roman font, not italicized*

**Authors Response**
- We have corrected the spelling for *S. meyeri*, and unitalicized v.

13. *Discussion – has there been similar detailed approaches taken on other long records? This seems to be a novel approach for considering the relations between diversity/ stability and climate, but that aspect is not highlighted by the authors and it should be! Has this approach of melding resource ecology and diversity been applied to other climate records, perhaps from varved lakes and accounting for Holocene scale records (LIA, MCA, HCM)?*

**Authors Response**
- As far as we are aware, there are few detailed records as ours, hence the novelty and value of this study. And there has not, as far as we are aware, been other studies which have looked at relations between productivity-diversity and climate over such long timescales. We think we

highlight this novelty in the Introduction Lines 109-113, "*This constitutes an important gap in our knowledge because in terms of climate change, PDR and resource use efficiency (Gross and Cardinale 2007; Ptacnik et al. 2008) will be fundamentally different over long (e.g. climate and landscape evolution) and short (e.g. pulse disturbances such as climate disturbance events (Kéfi et al. 2019)) timescales.*"

- We have rewritten and refocused the introduction quite considerably to emphasise and explain this part of the paper much more, and we emphasise the novelty and importance of this record again in the Discussion

14. *Discussion – earlier efforts by Khursevich et al. (2001, 2005) and Edlund (2006) have considered the longer Baikal records, but in lower resolution and with fewer measures of diversity and productivity. How does this new record compare or contrast with those earlier approaches to examine the full Baikal record. Is the Pleistocene/Holocene transition unique? For example, this paper suggests higher valve flux and BVAR in glacial times (T1) vs Holocene (see Table 1). This seems to contrast with many of the diatom depauperate regions characteristic of other glacial periods in Baikal's history*

**Authors Response**
- The reviewer makes a really useful suggestion in comparing our records to previous studies done on Lake Baikal, albeit at lower resolution. We have added a new section (see below) which brings in the three studies that they highlight, plus Bradbury et al. 1994, which covers a very similar timescale as our own, with cores from off the shoulder of the Selenga Delta spanning the past 15,000 years or so.

**New section added:**

4.4 Comparisons to other studies within Lake Baikal

15. *Discussion – the Baikal diatom community is characterized by high endemicity. Is there any reason to believe that this endemic flora drives the patterns shown in your data, i.e. is the resiliency a by-product of endemicity? ln 521-533*

**Authors Response**
- This is a very interesting question. Rather than the endemicity *per se,* it's likely the length of time spanned by the continuous sedimentary records found in ancient lakes that is the primary factor promoting resilience. Of course, as endemicity needs a lot of time to develop, ancient lakes have generally high endemicity. We have included this concept in our discussion, drawing in work from the recently published paper by Luethje & Snyder (2021, Phytotaxa) in which they discuss how climate events appear to drive morphological transition within a "species complex" (in that case *Pantocsekiella*, i.e. the species around the "*Cyclotella ocellata/comensis* complex) on a long time-scale (0 to 1.2 Ma) in Lake El'gygytgyn. In their study, they identify a morphodeme as a distinct species, partly on the basis that its occurrence in the record corresponds with stable climatic conditions during a 180 ka-long interglacial (when the lack of competition from other diatoms allowed the speciation of this large-sized morphodeme). The idea is, given sufficient time and opportunity (i.e. the demise of a competitor for a particular niche), morphological variants of a species (in their case, a large-celled variant) will be allow to thrive and occupy the vacant niche. The *Cyclotella (Lindavia) baicalensis/ornata/minuta* complex in Lake Baikal is another example of speciation as the different sizes allow them to occupy different habitat/seasonal niches.

16. *Minor corrections: ln 52, understanding: : :is ln 117, remove comma after ": : :2018) restricted: : :" ln 130, change to ": : :diversity that is not experienced: : :" ln 133, change to ": : :events disrupt these : : :" ln 140/41, odd expression outside of UK, change to ": : :due to its diverse*

*flora...” ln 142, provide reference to endemicity of Baikal ln 145, italicize the ship's name. She deserves that. ln 149-50, add space before meter abbreviation in 3 places. Check rest of msc for same. ln 160/61, superscript 14 in 14C, check rest of msc for same, also noted for _m3 (ln 215), etc. ln 222, clarify what is meant by PDR? Is PDR the relationship between paleoproductivity and N2 or is it N2? If it is the relationship, how is it calculated? Fig 5 seems to be reporting this "PDR" but PDR is not described or connected to Fig 5. What am I missing as a reader here? Is Fig 6b also related to PDR? ln 399, ": : :Bolling, the pre-Bolling diatom: : :"*

**Authors Response**
- We have taken all of these onboard

*17. ln 415, why is there a delay in the N2 diversity decline? Would be worth some speculation.*

**Authors Response**
- We actually speculate on this from lines 461-471… *"Diatom responses to climate change within the Younger Dryas were instantaneous but complex, kick-started by the first appearance and rapid growth of Ulnaria acus (Fig. 3) (indicative of higher dissolved silica concentrations in the water column (Bradbury et al. 1994)), and increasing abundance of C. inconspicua. However, the decline in N2 diversity only from c. 12.6 cal kyr BP (Fig. 4), suggests that resources for diatom growth were not initially limiting. Occupying different seasonal niches, spring blooming U. acus and autumnal blooming C. inconspicua (Ryves et al. 2003) are unlikely to be directly competing for the same resources, in part because small centric diatoms do not utilise a lot of silica (Bradbury et al. 1994). Having several co-dominant species is reflected in the relatively high evenness scores for the Younger Dryas (Fig. 4), related to few resources being limiting (Interlandi & Kilham 2001)."*

*18. Ln 456, skvortzowii misspelled ln 475, close parenthesis after 2011) ln 475, verb agreement "shows" ln 509, check msc for formatting of N2, N0, N1, italicized N and Roman 2 seems the standard. ln 541, close parenthesis after Fig. 3)*

**Authors Response**
- We have corrected these errors.

**Dr Gavin Simpson comments**

1.  *My first comment relates to the diversity measures and how they have been handled. Hill's numbers, like other count, or count-based, measures are affected by sampling effort; all else equal, the greater the sampling effort the greater the diversity, the greater the Hill number. In palaeolimnological studies, sampling effort concerns at least two elements of the data collection and analysis process: i) the sample count, here _300 valves; and ii) the sedimentation processes, accumulation rates, and depth-based sampling that generated the sediment slices that were counted. The authors appreciate these issues and employed methods to handle varying-effort source 1 via a bootstrap approach… It is unclear to what extent variation in accumulation rates might contribute to or obscure the underlying trends in diversity metrics in the core as accumulation rates are not presented alongside the time series of diversity metrics… A more complicated alternative might be to not use any rarefaction at all, and instead model the observed data using a regression approach, with an offset that includes both the count total and the number of years per slice. For the regression model itself, a GAM seems appropriate given the non-linear change in the metrics over time*.

**Authors Response**
-   Dr Simpson is essentially asking how can we account for variable SARs (which vary by a factor of 4) in interpreting Hill's numbers for species richness / diversity, and he helpfully provides a number of suggestions to consider. We agree and have reanalysed the data to take into account SARs when presenting and discussing our diversity measures.
-   Our study now focusses on new analyses for rarefied Hills diversity measures $N0$, $N1$ and $N2$ divided by the accumulation rate over time.
-   Note that we have also taken the opportunity to update our age modelling using IntCal20, although specific dates over our timeframe are similar to those obtained from using IntCal13.

-   We have also looked at Dr Simpson's suggestion of the more complicated alternative, calculating Hill's diversity measures without randomization and using the predicted values of a GAM model specified as gam.rich1 <- gam(N0 ~ s(age, k = 70), offset = log(counts*AccumRate), family = poisson, method = "REML", data = diversity). However, while we have undertaken these analyses, we settled on the simpler approach above.

2.  *The second area I wish to comment on is in the interpretation of the moving window coefficient of variation (CV) results, especially Figure 4. What is being shown in Figure 4? CV is a unit-less variable, but each plot has a label on the y-axis. In the upper panel we see variation in the CV but most of these changes are relatively small suggesting CVs of between 9 and <3 % of the mean. Are these values biologically meaningful? Is the fact that we see CV of log(BVAR) declining in the Late Glacial and early Holocene, really an important decline? The N2 signal seems to show more variation; apart from the Late Glacial period, CV values are between 10% and 50% of the mean N2. Those numbers seem, however, at first sight to indicate that the stability of N2 is much less than log(BVAR). My comment really relates to one of whether palaeo productivity stabilised, and if so, how does that mesh with the interpretation that N2 was also stable for much of the Holocene where the variability in CV values for N2 is so much higher than for BVAR?*

**Authors Response**
-   Dr Simpson's concerns here relate to (i) whether the interpretations we make with respect to changing CV over time are ecologically meaningful for log(BVAR) (as variation is rather minor) and has palaeoproductivity really stabilised during the Holocene; (ii) we originally interpreted N2 as being rather stable for the Holocene, yet higher variability for N2 is counter to that interpretation. We note that Dr Simpson didn't object to the use of CV, but had queries about interpretations surrounding stability.

- We agreed with these issues and, after refocusing the introduction focus on diversity and productivity relationships at different time periods (see below), we have decided to drop CV from the manuscript all together.

3. *The Kruskal Wallis analysis is non-parametric, but it is not without assumptions. The principal assumption that is certainly violated here is an assumption of independent observations, because your data are a time series. I don't know if there is a way to correct the p-values for the loss of degrees of freedom due to the dependence between samples (as is often done for regression, assuming AR noise), but if not, one might have to use a permutation or resampling/bootstrap procedure where the permutation or resampling is done in a manner that preserves for correlation structure (ruling out simple permutation and simple bootstrap resampling).*

**Authors Response**
- We agree with Dr Simpson's comments and have deleted this section.

4. *My final comment relates to the correlation analysis as shown in Figure 5. What type of correlation are you showing here? All the standard correlation coefficients are bounded -1, 1 but the plot shows values >1. I suspect what you're showing is the coefficients from a linear regression through the scatter of productivity and diversity data in the 1000 year moving windows? This needs to be fully explained in the methods. Putting an informative label on the y-axis would also help the reader understand this figure. Another problem with Figure 5 is the use of the p values encoded as colours on the plot. It's impossible to tell what the p-values are at the low end, where the interest lies, because of the colour scheme used. Also, you (or the reader) are at risk of making a massive multiple comparisons mistake here. If you are going to use the p values then you would need to correct them for multiple comparisons (i.e. for as many tests as the number of data points shown) using the false discovery rate to adjust the p values.*

5. *Then you could use a binary indicator perhaps to show which values remain significant. Other questions remain with this correlation analysis: is a linear model in a moving window a good fit to the data?; if relying on the p values, how biased are these because neither the response nor the covariate are independent; if this is a linear regression, which variable played the role of a dependent and independent variables?*

**Authors Response**
- These comments were perhaps the most substantial for us to deal with as they go to the heart of what we are trying to get out from our data. As pointed out, one of the problems of the moving window analysis was that it was highly subjective to the size of the window, and Dr Simpson was right in that sometimes a linear model might not be the best model to describe the data.

- We have decided instead to take a different approach to investigate productivity-diversity relationships over time, which is better driven by ecological understanding, and, we believe, to be more robust. Through independent palaeoclimate studies, we have identified well-known periods of global climate change, but also rooted in independent climate reconstructions from southern Siberia over the past 16,000 years.

- Using generalised additive models (GAMs), we test and compare productivity-diversity relationships identified in the table below. These are based on new hypothesis which are much more exploratory in nature, given that we don't have any expectations of how data should behave over such long timescales. Our new approach is highlighted in the introduction.

6. *Regarding the principal curves, how confident are you that the data are well represented by a single gradient? The eigenvalues of the CA that you used for the starting curve would be a guide as to whether there is remaining structure in the data on axes 2 and higher. PrCs are really useful when there is a strong single gradient in the data, but when there are secondary gradients my experience is that they are much less useful and can get stuck in some weird solutions that don't make ecological sense. You can get an idea of this by looking at the complexity of the smoothing splines fitted to individual taxa; if these are not simple linear, monotonic, or unimodal curves than it is more likely that the PrC is being asked to do too much and is a good sign of problems with the fit. I mention this because of the very rapid changes in the PrC scores, which can happen when the PrC itself is too complex; it would be useful for the reader to see the 2d ordination with the PrC superimposed on it, perhaps in the supplementary materials?*

**Authors Response**
- As suggested by Dr Simpson, we checked the result of the CA. The variation capture by axis 1 was only explaining ~4% of the total inertia/variation. The first four axes seem relevant (in total explaining a little bit more than 11% of the total variation), so based on this actually PrC turned out not be a very good choice in this case.

- Comments from Reviewer 1, Point #5 have already persuaded us that we needed to reanalyse our data based on separating out planktonic taxa from benthic taxa, as multiple gradients indicated by the PrCs may be, for example, reflective of the different habitats. We have chosen to keep things simple and show DCA axis 1 sample scores to give an idea of turnover over the time period. Influence of time is undertaken using CCA.

7. *Finally, throughout there are few attempts to quantify the uncertainties in the quantities you estimate, interpret, discuss, and present to the reader, or to compare the observed results with appropriate null models of no change/trend. This makes it difficult to gauge the overall support of your interpretation that comes from the data and the analyses*

**Authors Response**
- We think through a more focussed, hypotheses-driven approach our interpretations that come from the data and the analyses are more robust, and we ensure that uncertainties are considered throughout.

8. *Minor Comments- L189: there's an extra parenthesis before "De'Ath"; - L215: here and throughout, the superscripts in your units appear to have gone missing, perhaps during the conversion to PDF?; - L224: here and throughout, the way you present the Hill's numbers changes through out the manuscript. Sometimes the "N" is in italics, sometimes not, and sometimes the number is in italics and sometimes not; - L243: the R^2 should be upper case R and the 2 is not superscript; - L294: delete "as" between "declined" and "rapidly" or change the sentence to indicate what BVAR changed as rapidly as; - L501 add "zone" after "photic"*

**Authors Response**
- These have all been taken account of. For information, all Hill's numbers are given with $N$ italicised, and the following number not, i.e. $N$0, $N$1 and $N$2 (as per Felde et al. 2016).

**References used in this report, and now added to the manuscript**

Belykh, O. I., Ekaterina, G., Sorokovikova, T., SaphonovaI. A., Tikhonova, V: Autotrophic picoplankton of Lake Baikal: Composition, abundance and structure. Hydrobiol., 568, 9-17, 2006

Bradbury, J. P., Bezrukova, Ye. V., Chernyaeva, G. P., Colman, S. M., Khursevich, G., King, J. W., Likoshway, Ye. V.: A synthesis of post-glacial diatom records from Lake Baikal. J. Paleolim., 10, 213–252, 1994.

Edlund, M. B.: Persistent low diatom plankton diversity within the otherwise highly diverse Lake Baikal ecosystem. Nova Hedwigia 1310, 339–356, 2006.

Khursevich, G. K., Karabanov, E. B., Prokopenko, A. A., Williams, D. F., Kuzmin, M. I., Fedenya, S. A., Gvozdkov, A. A.: Insolation regime in Siberia as a major factor controlling diatom production in Lake Baikal during the past 800,000 years. Quat. Int. 80–81, 47–58, 2001.

Khursevich, G. K., Prokopenko, A. A., Fedenya, S. A., Tkachenko, L. I., Williams, D. F.: Diatom biostratigraphy of Lake Baikal during the past 1.25 Ma: new results from BDP-96-2 and BDP-99 drill cores. Quat. Int., 136, 95–104, 2005.

O'Donnell, D. R., Wilburn, P., Silow, E. A., Yampolsky, L. Y., Litchman, E: Nitrogen and phosphorus colimitation of phytoplankton in Lake Baikal: insights from a spatial survey and nutrient enrichment experiments. Limnol. Oceanogr. 62, 1383-1392, 2017.

Luethje, M., Snyder, J.,: Climate-related morphological changes in Pantocsekiella (Mediophyceae) spanning 0-1.2 Ma in the Lake El'gygytgyn, northeastern Russia including Pantocsekiella elgygytgynensis sp. nov. *Phytotaxa*, 478, 67-91, 2021.

Panizzo, V. N., Swann, G. E. A., Mackay, A. W., Vologina, E., Alleman, L., Andre, L., Pashley, V. H., Horstwood, M. S. A: Constraining modern day silicon cycling in Lake Baikal. Global. Biogeochem. Cy. 31, 556-574, 2017.

Popovskaya, G. I.: Ecological monitoring of phytoplankton in Lake Baikal. Aquat. Ecosyst. Health. 3, 215-225. 2000.

Popovskaya, G. I., Likhoshvai, E. V.: *Plankton Diatom Algae in Lake Baikal: Key Atlas*, Novosibirsk: Nauka, 2011. [in Russian].

Popovskaya, G. I. Usol'tseva, M. V. Domysheva, V. M. Sakirko, M. V. Blinov, V. V. Khodzher, T. V: The spring phytoplankton in the pelagic zone of Lake Baikal during 2007- 2011. Geogr. Nat. Resour. 36, 253-262, 2015.

Satoh, Y. Katano, T. Satoh, T, Mitamura, O. Anbutsu, K. Nakano, S. Ueno, H. Kihira, M. Drucker, V. Tanaka, Y. Mimura, T. Watanabe, Y. Sugiyama, M: Nutrient limitation of the primary production of phytoplankton in Lake Baikal. Limnology 7, 225-229, 2006.

Swann, G. E. A, Panizzo, V. N., Piccolroaz, S., Pashley, V., Horstwood, M. S. A., Roberts, S., Vologina, E., Piotrowska, N., Sturm, M., Zhdanoc, A., Granin, N., Normal, C., McGowan, S., Mackay, A. W: Changing nutrient cycling in Lake Baikal: the world's oldest lake. Proc. Nat. Acad. Sci., 117, 27211-27217, 2020.

---

## Author Response (AR2)

Dear Dr Mills,

Thank you for the comments from the reviewer for the revised version of our manuscript. And we're so pleased that they agreed that the revisions were all OK.

The corrections suggested be the reviewer and our responses are:
- Line 225-7: sentence needs to be rewritten: DONE
- Line 420: waters start to stratify: DONE
- Line 429: Ocean: DONE
- Line 446: generally increased P availability: DONE
- Line 448, 465: remove comma (don't split subject and verb with a single comma): DONE
- Line 502 and elsewhere: sometimes Si is used, sometimes dissolved silica, sometimes silica, etc – be consistent: MANUSCRIPT RE-READ AND CONSISTENT OF DISSOLVED SILICA USED WHERE RELEVANT
- Line 530: change on to in: DONE
- Line 547: comma after acus: DONE
- Line 554: change is to are: DONE
- Lines 561-2: Northern Hemisphere: DONE
- Line 564: remove comma: DONE

With best wishes,

Anson (on behalf of all the authors)